# Advances in the Development of Small Molecule Antivirals against Equine Encephalitic Viruses

**DOI:** 10.3390/v15020413

**Published:** 2023-02-01

**Authors:** Tyler J. Ogorek, Jennifer E. Golden

**Affiliations:** 1Department of Chemistry, University of Wisconsin-Madison, Madison, WI 53706, USA; 2Division of Pharmaceutical Sciences, School of Pharmacy, University of Wisconsin-Madison, Madison, WI 53705, USA

**Keywords:** alphavirus, antiviral, small molecule, VEEV, EEEV, WEEV, viral encephalitis, drug discovery

## Abstract

Venezuelan, western, and eastern equine encephalitic alphaviruses (VEEV, WEEV, and EEEV, respectively) are arboviruses that are highly pathogenic to equines and cause significant harm to infected humans. Currently, human alphavirus infection and the resulting diseases caused by them are unmitigated due to the absence of approved vaccines or therapeutics for general use. These circumstances, combined with the unpredictability of outbreaks—as exemplified by a 2019 EEE surge in the United States that claimed 19 patient lives—emphasize the risks posed by these viruses, especially for aerosolized VEEV and EEEV which are potential biothreats. Herein, small molecule inhibitors of VEEV, WEEV, and EEEV are reviewed that have been identified or advanced in the last five years since a comprehensive review was last performed. We organize structures according to host- versus virus-targeted mechanisms, highlight cellular and animal data that are milestones in the development pipeline, and provide a perspective on key considerations for the progression of compounds at early and later stages of advancement.

## 1. Introduction

The positive strand RNA alphaviruses, Eastern (EEEV), Western (WEEV), and Venezuelan Equine Encephalitis (VEEV) viruses are a subset of the *Togaviridae* family that cause significant disease in humans and equids. Viral transmission predominately occurs subcutaneously through the bite from infected mosquitoes, but other arthropods can also serve as vectors [1]. These “New World” alphaviruses, so termed owing to their historical localization in North, Central, and South America, present in humans with symptoms that range from a combination of fever, headache, muscle aches, and vomiting to a more serious state of infection, resulting in seizure, coma, or death [1,2,3,4]. The clinical manifestation of disease and long-term effects stemming from equine encephalitis virus (EEV) infection are distinguished from that of the “Old World” alphaviruses such as chikungunya (CHIKV), Sindbis (SINV), O’nyong-nyong (ONNV), and Ross River (RRV) viruses that generally cause fever, rash, and arthritogenic effects that can be long-lasting [5,6].

In humans, the incidence of natural VEEV infection is the highest of the three EEVs, resulting in hundreds of thousands of cases each year [7,8]; however, this is accompanied by the lowest case mortality rate of the three viruses at <1% [9]. Importantly, VEE symptoms are similar to that of other viral diseases such as Dengue fever, and in areas where both viruses circulate, it is estimated that ~10% of the Dengue fever cases are misdiagnosed and are actually caused by VEEV [8,9]. Compared to VEEV, WEEV infection occurs less frequently in humans but has a higher case mortality rate of 3–15% [10,11]. Natural enzootic EEEV infection in humans is relatively rare; however, there is a higher documented mortality rate of 30–50% [4]. In fact, the typical average of 11 EEE cases per year was dwarfed by the occurrence of 38 human cases in the United States during 2019 that resulted in the death of 19 patients [2,12,13]. Survivors of equine encephalitic infections frequently contend with persistent neurological complications [1,2,14,15], and these infections in the elderly and young children are associated with poorer outcomes [4,16]. As noted, infection can occur naturally through environmental vectors; however, for VEEV and EEEV, an intentional release of an aerosolized virus has been studied as a potential biowarfare agent [17,18,19].

Inactivated vaccines are available to protect equids, and either an inactivated or live, attenuated VEE TC83 vaccine has been provided for select laboratory and military personnel who risked exposure to these viruses; however, these vaccines are not suitable for use by the general public due to questions of efficacy and safety [20,21]. As such, there are currently no FDA-approved vaccines or therapeutics in hand for any alphavirus infection, leaving patients only with supportive care options [22] and creating a void of opportunity for these viruses to find advantage. Research efforts have focused on the development of vaccines [23,24,25], monoclonal antibodies [26,27,28], and small molecules as possible therapeutic or prophylactic measures. Having mechanistically divergent approaches available to address VEEV, WEEV, and EEEV is an attractive, and likely required, strategy to derive an antiviral approach that is independent of viral strain and infection stage, avoids the emergence of resistance, and is safe and effective for a patient population that is heterogeneous with respect to age, sex, and underlying medical conditions. Small molecules play an important role in the development of antiviral agents—as drugs themselves—or as chemical probes [29,30] that serve as a molecular lens through which novel aspects of virology are discovered, leading to intervention opportunities. Several published reviews describe small molecules that affect alphaviruses [17,31,32,33], including a comprehensive 2017 account that was contributed by Ching and co-workers [34].

This review focuses on small molecules that, since that 2017 report [34], have been newly discovered as VEEV, WEEV, or EEEV inhibitors or were previously reported, but have advanced to a new development milestone such as structure–activity relationship exploration, broader antiviral spectrum, mechanism of action studies, or have demonstrated efficacy in animals. Compound structures and associated data are generally organized by host-targeting or direct acting antivirals, and the mechanism underlying the compound activity is described when possible. When assessing the development landscape, it should be recognized that there is variability in assay conditions; therefore, comparing data across tabulated compounds should be done with caution due to differences in assay duration and endpoint, assay type and readout, viral strain, multiplicity of infection (MOI), and cell line, to name only a few with respect to in vitro assessments. For in vivo efficacy, several of the factors outlined for cell assays, along with the mouse strain and age, route and dose of viral challenge, timepoint in initiating treatment and its duration, compound route of administration and dose, formulation, and a myriad of compound related factors (e.g., solubility, stability, protein binding, tissue distribution, pharmacokinetics, etc.) play a role in the outcome of the evaluation. As such, the compilation of structures and data herein may be useful in selecting validated compound controls and aligning parameters for future experiments in this area.

## 2. Alphavirus Structure and Life Cycle

A single, positive-sense RNA strand contains the alphavirus genome. Within two open reading frames are encoded six structural proteins and four non-structural proteins (nsPs). A capsid protein (CP), envelope proteins (E1, E2, E3), a viroporin 6K protein, and a virulence factor known as the transframe (TF) protein [35] represent the structural proteins needed for cell entry, host defense mitigation, virulence, encapsidation, and budding events [36,37]. The non-structural proteins, nsP1, nsP2, nsP3, and nsP4, are involved in capping of RNA molecules, helicase and protease activity, avoiding host immune responses, processing of the viral polyproteins, and viral replication [38,39]. The viral RNA and capsid proteins form a nucleocapsid core that is surrounded by a protective lipid bilayer, or envelope, that is derived from the host plasma membrane during the budding and egress of newly formed virions [40]. The envelope displays glycoproteins essential to host cell receptor or attachment factor recognition, attachment, and entry (E2 glycoproteins) or post-entry membrane fusion (E1 glycoproteins) [41,42,43]. Generally, alphaviruses utilize clathrin-mediated endocytosis to infect host cells [44], though caveolae-mediated endocytosis [45] of Mayaro virus (MAYV), micropinocytosis [46] of CHIKV in human muscle cells, and pH-dependent pore formation with SINV [47] has also been described [48]. Once inside the host cell, the virus-containing endosome matures and undergoes a neutral to acidic pH shift, thereby leading to viral envelope/endosomal membrane fusion and release of the viral nucleocapsid into the cytosol [42,47]. Alphaviruses appropriate host cell proteins to translate the viral polyprotein p1234 which is further cleaved into individual nsPs. The nsPs have individual roles in the viral life cycle; however, an important function of these proteins involves their formation of a replication complex that facilitates negative strand RNA synthesis. Genomic and subgenomic RNA is subsequently synthesized, producing structural proteins that aid in virion assembly and release.

As with other viruses, small molecules have been discovered or designed to intervene in key events in the EEV life cycle. Generally, these can be categorized into two groups: those compounds that target essential, virus-recruited host proteins or compounds that directly engage viral proteins or processes. A compound that operates by a host-targeted mechanism has the potential benefit of working against a broader spectrum of viruses that all require the same host protein; however, this can lead to potential host related toxicity if the biological target is essential for host cell homeostasis and lacks protein or pathway redundancy. Compounds that target viral proteins can potentially avoid host cell toxicity as their mechanisms of action are independent of critical host proteins; however, viruses can evolve resistance mechanisms to overcome viral protein or pathway blockage, thereby reducing their utility. Of course, not all compounds suffer or are immune from these consequences in either category, and notably, as genomic and biochemical studies continue to elucidate how some small molecules operate, the lines between these groupings can blur. Nonetheless, the small molecules discussed in this review are presented with this classification in mind to aid the reader in navigating the small molecule EEV landscape.

## 3. Host Proteins and Immune Response as Antiviral Targets of Small Molecules

### 3.1. Host Interferon or Anti-Inflammatory Response as a Target

Infections from neuroinvasive VEEV, WEEV and EEEV can progress to encephalitis and lead to permanent neurological anomalies [14]. Damage results from the death of the neuronal cells that host the replicating virus or from a heightened inflammatory response that follows encephalitic alphavirus infection [49,50,51,52]. Not surprisingly, small molecule therapeutics have been sought to mitigate pro-inflammatory signaling or amplify cytokine activity that interferes with the viral life cycle. An advantage of this approach is that a precisely controlled, small molecule-driven stimulation of the host innate immune response may provide a broad spectrum and practical therapeutic strategy, especially for emerging novel pathogens for which antivirals have yet to be developed. In this section, recent advances in the identification and development of small molecules that modulate the host immune response in the context of encephalitic alphavirus infection are profiled (Figure 1 and Table 1).

Risner and co-workers evaluated the toxicity and efficacy of **celecoxib**, **rolipram**, and **tofacitinib**—three FDA-approved drugs with differentiated anti-inflammatory mechanisms of action—on human microglial (HMC3) and astrocyte (U87MG) cells that were infected with VEEV (Figure 1, Table 1) [53]. Cytotoxicity assessment of all three compounds in each cell type was performed, showing >90% HMC3 cell viability at a concentration of 50 μM, while U87MG cells could only tolerate a maximal concentration of 10 μM of each compound. A 100-fold reduction in VEEV TC83 titer was observed when infected HMC3 cells were pre-treated with celecoxib, while 10- and 5-fold viral titer reductions were observed for tofacitinib and rolipram, respectively. Efficacy in U87MG cells was diminished compared to what had been seen for the same compounds in HMC3 cells; however, the trend remained the same. Pretreating VEEV TrD-infected HMC3 cells with celecoxib reduced viral titer by 6.5-fold, while a 2-3-fold reduction was seen with tofacitinib and rolipram at the same concentration of 50 μM. Additional studies demonstrated that VEEV infection in HMC3 cells induced the upregulation of several pro-inflammatory cytokines while pretreatment of these cells with celecoxib (50 μM) reduced mRNA levels of several cytokines, suggesting that inflammation resulting from VEEV infection could be suppressed with celecoxib, a non-steroidal anti-inflammatory drug (NSAID) that inhibits the cyclooxygenase-2 (COX-2) enzyme that is important in pro-inflammatory pathways [53].

The immune response is regulated through the intermediacy of many signaling pathways and mediators which include the interferons (IFNs). Interferons are secreted by cells in response to pathogenic stimuli, thus inducing transcription and translation of effector proteins of the innate and adaptive host immune response [54,55,56]. Given that certain interferons can inhibit viral replication, selective and controlled upregulation of IFN has been explored as a potential means of thwarting viral infection [54,55,56]. In 2018, DeFilippis and co-workers reported the discovery of a sulfur-containing phenylacetamide, **C11** (Figure 1, Table 1) [57]. The compound resulted from a high throughput screen aimed at identifying small molecules that stimulated the type-1 IFN response, as measured through reporter proteins sensitive to IFN signaling in human telomerase-transduced human fibroblast (THF) cells [57]. A series of studies were conducted to triangulate which signaling proteins may be required for the observed C11-mediated type-1 IFN expression. These studies revealed a dependency on the Stimulator of Interferon Genes (STING) pathway which mediates immunological defenses during viral infection. Antiviral activity was surveyed across a panel of alphaviruses, including VEEV, CHIKV, ONNV, MAYV, and RRV. Compound C11 was tested against VEEV TC83 in THF cells without notable cytotoxicity, and showed a 4 log reduction of viral titer at a compound concentration of 50 μM. Ten analogs bearing various structural changes and all lacking the sulfur atom linker present in the parent compound failed to activate type-1 interferon signaling as observed with C11, highlighting the preliminary structural requirements of this scaffold as a STING-agonist prototype with anti-alphavirus activity [57].

**Tilorone** and **cridanimod** (Figure 1, Table 1) are among the earliest small molecules identified that potently induce IFN production in murine macrophages and mice through the intermediacy of the mSTING pathway [58]. While studies in human cells such as PBMCs, fibroblasts, and HEK293T cells, which notably have variable responses to IFN induction, revealed that these compounds did not similarly engage the hSTING pathway or produce a strong IFN response in human patients [58,59], literature documentation of in vitro and clinical antiviral efficacy abounds, and the compounds have been used clinically as antivirals in humans in various countries [60,61,62]. Recently, Keyer et al. studied these compounds in CD-1 mice and Wistar rats using a molecularly cloned VEEV variant, cTC83/TrD. The cTC83/TrD virus, so named because it contains two nucleotide substitutions from the TC83 genome, resulting in increased homology and virulence as that of wild-type Trinidad donkey (TrD) strain, was confirmed to induce a lethal cytopathic effect in BHK cells [63]. The strain was 100% lethal in untreated mice. Infected rats showed lethargy and reduced feeding, but all survived. Survival was determined for subcutaneously cTC83/TrD-infected mice that were treated with tilorone dihydrochloride salt or cridanimod meglumine salt throughout the duration of the 10-day experiment. All untreated control mice succumbed to infection, but 60% from each treatment group survived. Viremia was reduced by both compounds in rats and mice, but only in mice did the compounds induce IFN production. The studies demonstrated anti-VEEV efficacy for tilorone and cridanimod in mice and reduction of viremia in both rodent species; however, the absence of significant IFN induction in rats suggests that the observed antiviral effects are due to an IFN-independent mechanism [63]. While additional studies are needed to elucidate mechanistic details, these compounds have served as scaffolds upon which medicinal chemistry campaigns have been launched, seeking improved human STING pathway modulators [59].

The cyclohexane-linked dimer, **4210**, is a known inhibitor of myeloid differentiation primary response protein 88 (MyD88), a signaling adapter protein frequently upregulated during infection (Figure 1, Table 1). Previous studies had shown that 4210 reduced MyD88-dependent, pro-inflammatory signaling after bacterial toxin exposure, resulting in protection from lethal toxin challenge in mice [64]. Following reports that viral infections increase MyD88-dependent signaling [65,66] with concomitant impairment of type- 1 IFN antiviral activities [66,67], Saikh and co-workers showed that MyD88 inhibition by 4210 amplified IFN-beta production in human glioblastoma astrocytes (U87MG cells) in a dose dependent manner [68]. Antiviral activity was determined for 4210 with EEEV FL-93-939 or VEEV TrD, resulting in IC_50_ values of 11 and 33 µM, respectively, and a reduction in viral titer. Cytotoxicity in U87MG cells was not observed for tested concentrations of 4210. Examination of physiochemical and in vitro absorption-distribution-metabolism-excretion (ADME) parameters of 4210 revealed modest solubility (<50 μg/mL) and limited microsomal stability (t_1/2_ = 21 min). For mice challenged with VEEV TC83, a 30% improvement in survival compared to untreated controls was observed for pre-treated, infected C3H/HeN mice administered 4210 over 7 days. Compared to untreated mice in the study, mice treated with 4210 showed a reduced severity of disease based on clinical scoring of symptoms. Taken together, the outcomes of these studies show that inhibition of MyD88 by 4210 can amplify IFN production during viral infections and stands as another example of targeting host proteins to intervene in alphavirus infection.

### 3.2. Inhibitors of Nuclear Protein Transport

The trafficking of proteins in and out of the nucleus of an infected host cell is an important part of the viral life cycle that includes attenuation of the innate immune response [69,70]. Nuclear transit requires recognition by importin (IMP) transporters, such as the heterodimeric IMPα/β, that modulate binding and translocation of cargo through the nuclear envelope by way of an integrated nuclear pore complex (NPC). Another function attributed to the VEEV capsid protein, besides protecting newly generated viral RNA, is binding IMPα, thereby preventing heterodimer formation or impacting NPC transit directly [71,72]. As a result, the host cell immune response that relies on functional nuclear trafficking is impeded. Disruption of VEEV capsid–IMPα/β1 association is an antiviral strategy pursued by Jans et al. who employed an in silico, structure-based drug design screen to identify inhibitors of this specific protein–protein interaction [73,74,75]. Using a computational model of VEEV capsid–IMP interactions, a virtual screen was performed, followed by confirmation of hits in an Amplified Luminescent Proximity Homogenous Assay (AlphaScreen) that detects disruption of a protein–protein interaction. Compound **1111684** (Figure 2, Table 1) emerged with an IC_50_ value of ~5 μM and was separately found to inhibit the nuclear localization of the VEEV capsid protein. In Vero cells infected with VEEV TC83-luciferase, 1111684 showed an EC_50_ of 9.9 μM and a CC_50_ = 36.4 μM. At 10 μM, 1111684 reduced VEEV TC-83 titer compared to DMSO control by ~1 log [73]. The compound was inactive when deployed against a capsid protein-mutated strain of VEEV TC83 which was incapable of engaging IMPα, providing evidence that 1111684 disrupts the capsid–IMP interaction. In a follow-up AlphaScreen performed by this research group, several structurally related 1,4-diazepines bearing ring-fused pyrrole and tetrahydrobenzothiophene moieties were identified [76]. Of these, **G281-1485** and **G281-1564** inhibited VEEV capsid–IMPα/β1 interaction with IC_50_ values of 12.2 μM and 25.0 μM, respectively. In Vero cells infected with the reporter virus, VEEV TC83-luciferase, G281-1485 and G281-1564 were active without significant cytotoxicity. In Vero cell-based plaque assays, only G281-1564 was tested in this series, revealing a >95% reduction in viral plaque formation compared to DMSO control at a concentration of 50 μM and a 20% reduction at 10 μM. These and further studies [77] of G281-1564 showed that the compound interfered with host cell nuclear import of VEEV capsid protein, and inherent structure–activity relationships that emerged from the AlphaScreen suggested that potency and cytotoxicity may be modified with further medicinal chemistry effort.

Mifepristone is a synthetic steroid that, due to its antagonism of the progesterone receptor, is FDA-approved for medical abortion [78,79]. The drug has antiviral activity against HIV due to inhibition of importin α/β1 (IMPα/β1) binding to HIV integrase, leading to disrupted nuclear transport [74,75]. Nuclear transport of VEEV capsid protein is also impacted by mifepristone [80]. A medicinal chemistry effort was undertaken to assess if the antiviral effects of mifepristone could be optimized while avoiding the progesterone receptor antagonism responsible for the abortifacient effects [81]. SAR was focused on the A-ring carbonyl group and the C11- and C17-positions of the steroid framework. Most of the analogs were prepared by any one of several 3–5 step protocols which were employed to produce a set of 27 analogs. Mifepristone was modestly potent in a VEEV TC83 based luciferase reporter assay without notable cytotoxicity in Vero cells (EC_50_ = 19.9 μM, CC_50_ = 165 μM). Modifications at the C17-position revealed a preference for a hydroxyl group at that position paired with a hydrophobic alkyne bearing a terminal trialkylsilyl substituent. This C17 modification was integrated for structural changes surveyed at C11, resulting in several analogs with single digit micromolar VEEV EC_50_ values. The A-ring ketone or its replacement by a 1,3-dioxolane was beneficial, depending on the substitution patterns at C17 and C11. VEEV titer was assessed for a subset of compounds at 10 μM, revealing structure dependent percent titer reductions ranging from 47–98% [81]. Select analogs were found to inhibit VEEV capsid protein nuclear import and of these, **compound 50** (Figure 2, Table 1) did not antagonize the progesterone receptor as indicated by monitoring effects on gene transcription in the presence of a competitive agonist. An 11-fold improvement in VEEV TC83-luc inhibition and no noticeable cytotoxicity was observed with compound 50, and an 86% reduction in viral titer was demonstrated at a concentration of 10 μM. Docking studies between select analogs and the progesterone receptor were performed to account for compound profile differences and provided insights into future structural modifications [81].

### 3.3. Modulators of Host Protein Phosphorylation

Kinases are central to a variety of cellular processes and affect functional changes through phosphorylation of their target protein or substrate. As viruses depend on the intermediacy of host cell proteins, the selective inhibition of key kinases has drawn immense interest in the quest for potentially broad-spectrum antiviral agents. Given that kinase inhibition has been a successful strategy in treating various diseases, the repurposing of kinase inhibiting preclinical compounds or FDA-approved drugs has been examined as a means of identifying antiviral therapies or opportunities for intervention [82,83,84]. In this review, several kinase inhibitors or compounds that modulate host protein phosphorylation are highlighted that were assessed against encephalitic alphaviruses.

The pyrrolopyrimidine, **R10015** (Figure 3, Table 1), was identified by Yi et al. as a LIM domain kinase 1 (LIMK1) inhibitor that blocked HIV-1 in cells [85]. As kinases such as LIMK are integral to cytoskeletal organization and dynamics, they can be recruited during viral infection to facilitate viral entry and egress of newly assembled virions [86,87,88]. R10015, reportedly a binder of the ATP binding site, showed potent biochemical inhibition of human LIMK1 (IC_50_ = 0.038 μM) and a cell shifted EC_50_ of 14.9 μM in HIV-1 infected cells. In an assessment of antiviral spectrum, R10015 inhibited luciferase tagged VEEV TC83 in Vero cells with an IC_50_ of 5 μM and a 2 log reduction of VEEV TrD titer at a compound concentration of 50 μM in a separate plaque assay. While the report centered on studies involving HIV-1, the identification of a host-dependent kinase that appears to be important to multiple stages of viral infection may hold promise for the development of broad-spectrum antivirals from this structural class with improved VEEV efficacy.

**Sorafenib** is a drug used in the treatment of certain cancers [89,90] and operates through the inhibition of various kinases that are important to cancer cell proliferation and angiogenesis [91]. The tosylate form of the drug was identified in an antiviral screen performed by Lundberg et al. focused on repurposing FDA-approved drugs as host-targeting alphavirus inhibitors (Figure 3, Table 1) [92]. Sorafenib inhibited VEEV TC83-luciferase or VEEV TC83 with EC_50_ values <5 μM without discernable toxicity. Other alphaviruses were similarly or more potently inhibited, including VEEV ZPC738 (EC_50_ = 6.2 µM), EEEV FL93-939 (EC_50_ = 6.7 µM), SINV EgAr (EC_50_ = 1.3 µM), and CHIKV 181/25 (EC_50_ = 0.2 µM). Further studies were conducted to determine if the observed antiviral activity was due to Raf kinase inhibition using a structurally related sorafenib analog, SC1, which does not inhibit Raf kinases [93,94]. Both compounds reduced VEEV TC83 titers in cells (2 log for sorafenib) at a concentration of 10 μM, and silencing RNA experiments targeting B-Raf and C-Raf kinases ruled out their involvement in the observed antiviral effects. Additional mechanism of action studies revealed that sorafenib treatment reduced amounts of key phosphorylated proteins such as the mRNA cap binding protein, eukaryotic initiation factor E (eIF4E), and ribosomal protein S6 kinase (p70S6K). These outcomes occurred with concomitant reduction in VEEV capsid protein translation. Given that sorafenib has been shown in this and other studies [95,96,97] to inhibit multiple viruses through various means of impairing viral egress, the mechanistic underpinnings of the observed antiviral effects of sorafenib may hold a key to developing broad spectrum antivirals with improved profiles with this mode of action.

**Resveratrol**, a polyphenolic stilbene-based natural product that is known for its antioxidant properties (Figure 3, Table 1), was examined by Lehman et al. for possible anti-VEEV activity [98] due to its known inhibition of protein kinase B (AKT) and glycogen synthase kinase-3 (GSK-3), host proteins that reportedly facilitate alphavirus replication [99,100]. Using a luciferase reporter virus, VEEV TC83-luc, EC_50_ values for resveratrol of 21.8 and 20.1 μM were determined in Vero and U87MG cells, respectively, without notable toxicity. At a concentration of 150 μM, a 2 log reduction in viral titer was observed. **Pterostilbene** and **piceatannol**, stilbene analogs of resveratrol, demonstrated similar anti-VEEV activity in Vero cells infected with TC83-luc (EC_50_ ~29 μM). Resveratrol was found to reduce viral attachment and entry into host cells. Molecular docking studies reflected high binding affinities of resveratrol with the E1 and E2 envelope glycoproteins, results that support the observed effects on the early stages of infection; however, various compound concentrations did not alter viral titer or the percentage of infected cells at a given timepoint, suggesting that the observed antiviral effects are due to a more significant mechanism of action. In early stages of infection, resveratrol treatment reduced the amount of phosphorylation of proteins in the AKT pathway (AKT, GSK-3α, GSK-3β). Broader screening of resveratrol also revealed activity against SINV and CHIKV. Physiochemical and in vivo stability and metabolic liabilities likely limit the practical utility of resveratrol itself as an anti-VEEV agent in humans, and the reported activity of resveratrol across multiple targets [101] may obstruct a clear understanding of how these types of scaffolds work against VEEV. Nonetheless, these studies may lead to novel structures that assist in dissecting the underlying pathways that may play a role in the observed antiviral effects by resveratrol and related structures.

In vitro VEEV infection is accompanied by the activation of inhibitor kappa beta kinases (IKKs), a complex formed between IKKα, IKKβ, and IKKγ, that is responsive to cytokines and stress signals, and which regulates the nuclear factor kappa beta (NF-kB) pathway [102]. Viruses can intervene in this cascade [103,104,105], thereby evading the host immune response, and studies have linked IKKβ inhibition with a reduction in viral replication [106]. Using acrylonitrile **BAY-11-7082**, a known inhibitor of IKKβ, Bakovic et al. showed that IKKβ is responsible for the phosphorylation of specific sites on VEEV nsP3 which, according to mutation data, are necessary for the biosynthesis of negative strand viral RNA [107]. As understanding improves on which host–viral protein interactions are critical in the virus life cycle, small molecule disrupters of these processes can be potentially developed for therapeutic gain.

### 3.4. Inhibitors of RVxF, a Binding Motif of Protein Phosphatase 1α

The cyclopentane-fused quinoline, **1E7-03** (Figure 4, Table 1), is known to bind RVxF, a catalytic subunit within protein phosphatase 1α (PP1α), resulting in antiviral activity for HIV and Ebola virus [108,109]. Using 1E7-03, studies were conducted to assess the role of PP1α on VEEV replication [110]. In a VEEV TC83-luc assay, 1E7-03 showed an EC_50_ of 0.58 μM without significant cytotoxicity, and viral growth was decreased by >3 log when cells were exposed to a 10 μM concentration of 1E7-03. Viral titers were reduced in plaque assays using a 10 μM concentration of 1E7-03 with VEEV TrD and TC83 (1.5 and 2 log, respectively), WEEV 1930 California (3 log), EEEV GA97 (1 log), and CHIKV 181/25 (nearly 4 log). A series of experiments established that PP1α interacts with VEEV capsid protein, thereby altering the capsid’s phosphorylation state. Exposure to 1E7-03 resulted in changes to capsid phosphorylation at various sites, and this was linked to reduced binding of viral RNA to viral capsid protein. These insights informed the proposal of a mechanism accounting for a role of PP1α in the VEEV life cycle by which disrupted PP1α-mediated dephosphorylation of viral capsid prevents viral RNA binding and subsequent viral assembly [110]. These studies expanded the spectrum of viruses whose replication is affected by inhibition of the RVxF binding motif of PP1α. Further, inhibition of PP1α in this manner alters the phosphorylation state of VEEV capsid protein which impedes VEEV replication [110].

### 3.5. Inhibitors of Host Protein GRP78

**HA15** is a thiazole benzenesulfonamide that has been extensively studied due to its inhibition of glucose-regulated protein 78 (GRP78), a stress-induced, molecular chaperone involved in the unfolded protein response (UPR) of cells (Figure 5, Table 1). GRP78 can play a significant role in the development of envelope proteins of some viruses [111,112,113,114] and is overexpressed in cancer cells as a consequence of the accumulation of misfolded proteins [115,116]. Using an epitope tagged VEEV TC83 E2 protein, Barrera et al. screened for possible host proteins that engage the VEEV E2 glycoprotein through immunoprecipitation and proteomic analysis by mass spectroscopy [117]. These studies showed that HA15 reduced VEEV TC83 titer by 3 log at 50 µM. At this same concentration, HA15 showed a reduction of titer by 3-4 log for VEEV TrD, EEEV FL93-939, SINV EgAr 339, and CHIKV 181/25. By comparing intracellular and extracellular viral RNA levels, HA15 was shown to not inhibit viral RNA production, but rather inhibited viral proliferation at some step thereafter, such as viral assembly or protein trafficking.

### 3.6. Inhibitor of Host Mitochondrial Electron Transport and Pyrimidine Biosynthesis

**Antimycin A1a** is a secondary metabolite obtained from the bacterial strain, *Streptomyces kaviengensis*, that has a record of antiviral and antifungal activity (Figure 6, Table 1) [118,119,120]. Following the isolation of marine sediment, fermentation of actinomycetes, fractionation, and phenotypic screening in a cell-based WEEV replicon assay, Raveh et al. isolated and purified antimycin A1a as a hit compound [121]. Validation of antimycin A1a in a WEEV CPE assay determined potent activity. WEEV titer reduction was assessed across a panel of cell types at a concentration of 200 nM, revealing suppression of WEEV replication that was cell-type dependent. Several other structural analogs were assessed and compared with the A1a derivative, though the title compound showed the most promise with respect to potency. Evaluation of antimycin A1a in a mouse model of WEEV Cba87 infection showed that a twice daily 0.2 mg/kg dose of antimycin for 7 days reduced brain viral titer by 10-fold and led to a 40% survival after 14 days post-infection compared to the untreated control group. Antimycin A1a has a narrow therapeutic index [121,122]; however, its inhibition of host cell mitochondrial electron transport chain and pyrimidine synthesis is notable in the context of WEEV infection given that so few compounds have been characterized with efficacy against the virus in vivo.

**Figure 6 viruses-15-00413-f006:**
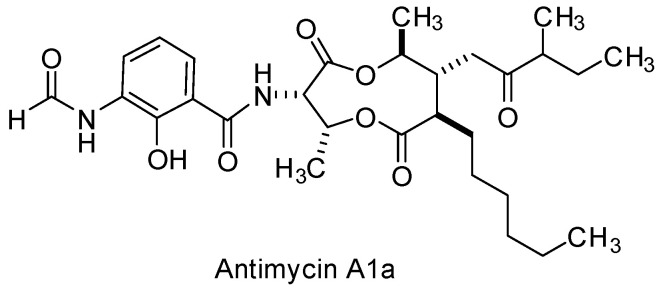
Chemical structure of antimycin A1a, an inhibitor of host mitochondrial electron transport and pyrimidine biosynthesis.

**Table 1 viruses-15-00413-t001:** In vitro and in vivo anti-EEV data for small molecules that target host proteins.

Compound	Virus Strain	In Vitro Antiviral Data					In Vivo Antiviral Data		Ref.
		Cell Line	MOI	EC_50_ μM	CC_50_ μM	Titer Reduction (Concentration)	Mouse *Challenge*	Survival *Dose*	
celecoxib	VEEV TC83	HMC3	0.1	-	>50 ^a^	100-fold (50 µM)	*-*	*-*	[53]
	VEEV TC83	U87MG	0.1	-	>50 ^a^	42% (10 µM)	*-*	*-*	[53]
	VEEV TrD	HMC3	0.1	-	>50 ^a^	6.45-fold (50 µM)	*-*	*-*	[53]
rolipram	VEEV TC83	HMC3	0.1	-	>50 ^a^	5-fold (50 µM)	*-*	*-*	[53]
	VEEV TC83	U87MG	0.1	-	>50 ^a^	23% (10 µM)	*-*	*-*	[53]
	VEEV TrD	HMC3	0.1	-	>50 ^a^	2-3-fold (50 µM)	*-*	*-*	[53]
tofacitinib	VEEV TC83	HMC3	0.1	-	>50 ^a^	10-fold (50 µM)	*-*	*-*	[53]
	VEEV TC83	U87MG	0.1	-	>50 ^a^	38% (10 µM)	*-*	*-*	[53]
	VEEV TrD	HMC3	0.1	-	>50 ^a^	2-3-fold (50 µM)	*-*	*-*	[53]
C11	VEEV TC83	THF	1	16.7 ^b^	>50	4 log (50 µM)	*-*	*-*	[57]
tilorone 2HCl	VEEV cloned TC83/TrD	BHK21 ^c^	-	-	-	-	CD-1 *SC challenge * *10LD_50_*	60% ^d^, *25.6 mg/kg/d, QD, 10 d, IG*	[63]
cridanimod meglumine	VEEV cloned TC83/TrD	BHK21 ^c^	-	-	-	-	CD-1 *SC challenge* *10LD_50_*	60% ^d^, *25.6 mg/kg/d, QD, 10 d, IG*	[63]
4210	VEEV TC83	Vero	10	24	>100 (A549 cells)	-	C3H/HeN *intranasal* *2 × 10^7^ PFU*	100% ^e^, *0.2 mg/kg * *BID, 7 d, IP*	[68]
	VEEV TrD	U87MG	10	33		3-4 log (33 µM)	-	-	[68]
	EEEV FL93-939	U87MG	10	11		3-4 log (11 µM)	-	-	[68]
1111684	VEEV TC83	Vero	0.1	9.9	36.4	1 log (10 µM)	-	-	[73]
G281-1485	VEEV TC83luc	Vero	0.1	7.5	>55	-	-	-	[76]
G281-1564	VEEV TC83luc	Vero	0.1	10.8	>55	>95% (50 µM)	-	-	[76]
compound 50	VEEV TC83luc	Vero	0.1	7.2	>100	>86% (10 µM)	-	-	[81]
R10015	VEEV TC83luc	Vero	0.1	~5	>100 ^f^	-	-	-	[85]
	VEEV TC83	Vero	0.1	~5	>100 ^f^	3 log (50 µM)	-	-	[85]
	VEEV TrD	Vero	0.1	~5	>100 ^f^	2 log (50 µM)	-	-	[85]
sorafenib tosylate	VEEV TC83luc	Vero	1	4.2	>80	-	-	-	[92]
	VEEV TC83	Vero	1	3.7	>80	2 log (10 µM) ^g^	-	-	[92]
	VEEV TrD	Vero	0.1	-	-	~2 log (20 µM)	-	-	[92]
	VEEV ZPC738 nluc	Vero	1	6.2	>80	-	-	-	[92]
	EEEV FL93-939 nluc	Vero	1	6.7	>80	-	-	-	[92]
resveratrol	VEEV TC83luc	Vero	0.1	21.8	314.5	2 log (150 µM)	-	-	[98]
pterostilbene	VEEV TC83luc	Vero	0.1	29.2	59.9	-	-	-	[98]
piceatannol	VEEV TC83luc	Vero	0.1	29.8	>500	-	-	-	[98]
BAY-11-7082	VEEV TC83	U87MG	0.1	-	-	>3 log (1 µM)	-	-	[107]
1E7-03	VEEV TC83luc	Vero	0.1	0.58	>100	>3 log (10 µM)	-	-	[110]
	VEEV TC83	Vero	0.1	-	>100	2 log (10 µM)	-	-	[110]
	VEEV TrD	Vero	0.1	-	>100	1.5 log (10 µM)	-	-	[110]
	WEEV California	Vero	0.1	-	>100	3 log (10 µM)	-	-	[110]
	EEEV GA97	Vero	0.1	-	>100	1 log (10 µM)	-	-	[110]
HA15	VEEV TC83	Vero	0.1	-	>200	3 log (50 µM)	-	-	[117]
	VEEV TrD	Vero	0.1	-	>200	3 log (50 µM)	-	-	[117]
	EEEV FL93-939	Vero	0.1	-	>200	3 log (50 µM)	-	-	[117]
antimycin A1a	WEEV Cba87	BE(2)-C	10	0.003	>1000	14-fold (200 nM)	C57BL/6 *SC challenge* *1 × 10^3^ PFU*	40% ^d^, *25* mg/kg *BID*, *7 d*, *IP*	[121]

EC_50_, effective concentration required to inhibit 50% of the treated cell population; CC_50_, cytotoxic concentration that results in death of 50% of the treated cell population; QD, once daily; BID, twice daily; SC, subcutaneous; IP, intraperitoneal administration; IG, intragastric administration; ^a^ data inferred from published figures; ^b^ EC_90_ value; ^c^ Used for VEEV clone validation; ^d^ compared to 0% survival in untreated controls; ^e^ survival in placebo or untreated controls was 50% and 70%, respectively; ^f^ Assessed in Rev-CEM-GFP-Luc cells; ^g^ MOI = 0.1.

## 4. Direct Targeting of EEV Non-Structural Proteins (nsPs) with Small Molecules

### 4.1. Non-Structural Protein 1 as a Target

As alphaviral mRNA is transcribed, the non-structural protein 1 (nsP1) facilitates the biosynthesis of a 7-methylguanosine nucleotide (m^7^-G), tethered by a triphosphate moiety (ppp, or TP), to the 5′ end of the mRNA strand. This 5′ cap prevents cellular enzyme-driven RNA degradation and aids in host immune response evasion, thereby blunting barriers to viral protein synthesis [123,124,125]. The 5′ cap is generated through a series of chemical reactions. This includes nsP2-mediated cleavage of the 5′ end terminal gamma-phosphate by RNA triphophatase (RTPase). The nsP1 mediates the N-methylation of GTP via a methyltransferase [MTase reaction] to generate m^7^-GTP. This is followed by guanylylation of nsP1 to form an nsP1-m^7^-GMP adduct [GT reaction] between viral nsP1 and m^7^-GTP with concomitant extrusion of inorganic pyrophosphate [123,126]. Transfer of the m^7^-GMP moiety to the 5′-diphosphate appendage of the RNA affords the capped strand, m^7^G(5′)ppp(5′)RNA. Given the importance of this process to viral replication, understanding of alphavirus-specific capping mechanisms and identification of inhibitors has been explored to design new antiviral agents [127,128]. For instance, Ferreira-Ramos and co-workers [124] developed a high-throughput ELISA assay based on the detection of the VEEV nsP1-m^7^-GMP adduct to identify new inhibitors of the GT reaction. A screen of 1220 Prestwick library compounds in the ELISA assay afforded hits that were further evaluated in a Western blot (WB) assay that quantified guanylylated VEEV nsP1. **Pyrimethamine**, a folic acid antagonist commonly used for chemoprophylaxis of malaria or to treat toxoplasmosis infection, emerged as an inhibitor of the VEEV GT reaction from the WB assay with an IC_50_ = 2.7 μM (Figure 7, Table 2). A set of seven structurally related 2,4-diaminopyrimidines were less potent by a factor of at least 6-fold with several analogs not showing inhibition of the GT reaction by WB analysis (>200 μM). The hydrated tartrate salt of **ketanserin**, an antihypertensive serotonin receptor antagonist featuring a quinazoline-2,4-dione core, was also identified by the WB assay as a GT reaction inhibitor, albeit with weaker potency than observed for pyrimethamine (IC_50_ = 14.6 μM). The pyridopyrimidin-4-one, **pirenpirone**, was also identified as a hit compound (IC_50_ = 39.6 μM) and, like ketanserin, is a known serotonin receptor antagonist that contains a 4-fluorobenzoylpiperidine moiety linked by two methylene units to a core nitrogen atom. Of the various commercial fragments and analogs of these hits that were assessed, only **altanserin**—a 2-thioxo derivative of ketanserin—showed comparable GT reaction inhibition (IC_50_ = 9.3 μM). Several compounds were also assessed separately for inhibition of the nsP1-mediated MTase reaction. At a concentration of 50 μM, pyrimethamine and ketanserin showed significant inhibition (74–79%) of the VEEV nsP1 MTase reaction while pirenperone showed less inhibition (45%). All three of these compounds demonstrated little to no inhibition of a human methyltransferase at that same concentration. Several known GT reaction inhibitors were used as assay controls, including triazolopyrimidinones MADTP-393 and MADTP-314 [129]. Several MADTP series compounds lost their inhibitory effect on VEEV nsP1 with the introduction of a D34S mutation; however, pyrimethamine and pirenperone both retained the ability to inhibit GTase activity with this variant, thereby suggesting that these compounds inhibit or bind VEEV nsP1 differentially compared to that of the MADTP series.

### 4.2. Non-Structural Protein 2 as a Target

The alphavirus nonstructural protein 2 (nsP2) is responsible for several key functions in the viral life cycle, making it an attractive therapeutic target [130,131,132,133]. Enzymatic activities accomplished by nsP2 include ATPase and GTPase activities [134] that are attributed to an N-terminal domain, helicase-mediated remodeling of viral RNA [135], proteolytic cleavage of the nonstructural polyprotein due to a cysteine protease domain [136], and RNA 5′-triphosphatase activity via an S-adenosyl-L-methionine-dependent RNA methyltransferase (SAM MTase) [137]. An X-ray crystal structure of the VEEV nsP2 protease has been resolved (5EZQ) [138], and a co-crystal structure with a protease inhibitor, E64d, has also been reported (PDB 5EZS) [139].

Zhang and co-workers [140] conducted a phenotypic screen using VEEV IC-SH3, a virulent strain isolated from humans during a VEE outbreak in the early 1990s, and a library of potential covalent binding compounds that might be expected to target the cysteine protease of VEEV nsP2. Based on an acrylate-linked 1,2-dihydroquinoline hit compound, a supporting cast of 15 analogs were generated to explore SAR in multiple cell types and with additional strains of VEEV. Key structural modifications of the hit scaffold included exchange of the acrylate warhead with an α,β-unsaturated methyl sulfone and saturation of the fused piperidine ring of the quinoline core. These efforts afforded **compound 11** which addressed hydrolysis of the acrylate warhead and revealed VEEV EC_50_ values that were consistently around 2 µM across multiple cell lines using TC83 and TrD strains (Figure 8, Table 3). Some of the most active compounds from the study, including compound 11, were evaluated for their ability to inhibit the VEEV nsP2-dependent proteolytic cleavage of a FRET substrate in vitro. Complete inhibition of the VEEV nsP2 proteolytic activity was observed for these compounds at a concentration of 20 µM, while at 0.9 µM, compound 11 inhibited 39% of VEEV nsP2 proteolytic activity after 24 h, and related analog, **compound 13**, demonstrated an inhibition of 71% at that same concentration. In silico molecular docking studies were conducted with compound 11 to provide a rationale for the covalent inhibitor binding and key interactions with nsP2. The authors proposed future studies to improve metabolic stability and potency while also assessing the pharmacokinetic suitability of this dihydroquinoline scaffold ahead of in vivo efficacy determination [140].

Haese and co-workers reported the identification and SAR exploration of a 2-quinolinone hit compound following a CPE-based phenotypic screen using VEEV TC83 in Vero cells, followed by hit validation using normal human dermal fibroblasts (NHDFs) [141]. These efforts revealed **SRI-33394** as a prime starting point for a systematic SAR investigation (Figure 8, Table 3). The team prepared approximately 81 analogs that surveyed five regions of the scaffold. Thiourea replacement with a urea was not tolerated, resulting in loss of potency; therefore, the thiourea was conserved. The presence of a basic nitrogen-containing functional group appended to the thiourea, along with a thiourea linked 2-methylfuranyl group, were also determined to be essential. Methylation of the SRI-33394 NH-quinazolinone core generated **SRI-34329** which showed a 6-fold improvement in CPE antiviral activity but a 3 log loss in viral titer reduction compared to the parent hit. Plaque assays and northern blot assessments showed that quinolinone SRI-34329 inhibited an early viral replication step that affected viral RNA synthesis. Resistance mutations from passaged VEEV TC83 in the presence of SRI-34329 were sequenced, showing Y102C or Y102S mutations in nsP2. Additionally, the compound improved upon its antiviral spectrum compared to the hit compound, revealing low micromolar activity against ONNV, MAYV, and RRV. Ultimately, liabilities in solubility and microsomal stability dominated and could not be appreciably addressed through structural augmentation in the analog set; however, these compounds and the collective studies provided insight into potential VEEV nsP2 inhibition and a rationale as to why select compounds were not active against CHIKV. In silico docking of the analogs was performed using the crystal structure of CHIKV nsP2 for which VEEV nsP2 has high homology. Though select compounds did not inhibit CHIKV, the CHIKV nsP2 contains a lysine residue at position 102 in contrast to a tyrosine residue in VEEV that, when mutated at this position, rendered VEEV resistant to tested compounds in the set. These models, in tandem with resistance mutant analyses and reverse genetics, were used to rationalize the observed SAR for the series and provide evidence of nsP2 as the target of these compounds [141].

### 4.3. Non-Structural Protein 3–Host Protein Interactions as a Target

The alphavirus non-structural protein 3 (nsP3) is an integral component of the viral replication complex which is responsible for generating genomic, subgenomic, and negative-sense strands of viral RNA during infection [142,143,144]; however, physical attributes of its disordered C-terminal domain, nsP3 localization beyond that of the replication machinery, and association with different host proteins implicate additional functions that facilitate viral replication [145,146,147,148,149]. As such, identification of host proteins that interact with nsP3 may provide intervention points and therapeutic opportunities. While some VEEV nsP3–host protein interactions had been previously identified [102,107,144,150], a recent study sought to broaden the interactome while also identifying small molecule-based inhibitors. Bakovic and co-workers examined host protein interactions with VEEV nsP3 using an HA-tagged nsP3 mutant protein and mass spectroscopy and immunoprecipitation assays [151]. The study revealed 160 VEEV nsP3–host protein interactions and 42 potential inhibitors, of which nine compounds were selected for further evaluation. In vitro VEEV TC83 and cytotoxicity assays highlighted **tomatidine**, **Z-VEID-FMK**, and the selective serotonin reuptake inhibitor antidepressant, **citalopram HBr**, at 10 μM as inhibitors showing a >10-fold reduction in VEEV titer (Figure 9, Table 3). Subsequent testing of these compounds at 20 μM also showed antiviral activity against VEEV TrD and EEEV. Knockdown studies employing short-interfering RNA (siRNA) and VEEV TC83 stop codon mutants were conducted to confirm the relevance of host proteins putatively targeted by the inhibitors. Eukaryotic initiation factor 2 subunit 2 (eIF2S2), the putative target of tomatidine, was determined to be important to VEEV genomic RNA synthesis and independent of VEEV subgenomic RNA generation. Tomatidine only slightly impeded VEEV genomic RNA translation and more significantly abrogated viral subgenomic RNA synthesis, suggesting possible alternative mechanisms in play. Transcription factor AP-2 alpha (TFAP2A), putatively targeted by Z-VEID-FMK and citalopram HBr, was shown to not be directly involved in viral RNA synthesis, despite inhibitors of this protein still displaying antiviral activity [151].

### 4.4. Non-Structural Protein 4 as a Target

Alphavirus nonstructural protein 4 (nsP4) is a component of the viral replication complex that harbors RNA-dependent RNA polymerase (RdRp) and terminal adenylyltransferase (TAT) activities [152,153,154]. Due to these pivotal roles in viral replication and a high level of conservation across viral RdRps [155,156], its inhibition or exploitation is an attractive strategy for antiviral development, as the compounds discussed in this section exemplify.

**Beta-d-N4–hydroxycytidine (NHC)** is a ribonucleoside mimic featuring an N-4-hydroxyl group on a cytidine core and is the pre-phosphorylated active form of the ester prodrug, Molnupiravir, which can be used to treat patients infected with SARS-CoV-2. A unique architectural feature of NHC (Figure 10, Table 3) is that it can exist in tautomeric forms, thereby mimicking cytidine in the hydroxylamine form, or in the oxime tautomeric form, it better resembles uridine [157]. As a consequence, once NHC is converted into its triphosphate metabolite (NHC-TP) and incorporated by the viral RdRp into newly transcribed viral RNA, the NHC-containing template is read inconsistently as either uridine or cytidine, generating mutations that impair the viability of the virus [157,158]. Further, the mutations are not recognized by proofreading viral exonucleases as erroneous, thus the misinformation within the genome is propagated. NHC was evaluated against smallpox in the 1970s and was later found to be active against hepatitis viruses [159], norovirus [160], and chikungunya virus [161]. As a broad-spectrum antiviral agent, Urakova and co-workers [158] evaluated NHC against VEEV TC83, showing submicromolar potency without discernable cytotoxicity. The studies revealed that NHC treatment was most effective when it was applied in the first 4 h of infection, and the VEE virions produced from NHC-treated cells contained mutations that compromised replication capability. Monitoring of emerging resistance after NHC treatment showed that resistance was challenging to achieve after even 20 passages and appeared to require multiple coincident mutations in nsP4 [158]. Additional studies in mice were undertaken to assess the efficacy and safety profiles of NHC, later designated **EIDD-1931** in preclinical development (Figure 10, Table 3) [162]. Painter and co-workers established the in vivo murine pharmacokinetic profile of EIDD-1931 and the triphosphate NHC derivative, **EIDD-2061**, after oral dosing, revealing distribution and exposure in plasma, spleen, and brain tissues and a good safety margin using up to 1000 mg/kg/day after repeat dosing for 7 days. Intranasally exposed, VEEV TrD-infected mice were used to assess the in vivo efficacy of EIDD-1931. Mice treated prior to viral exposure showed 90% survival in a 14-day study when dosed orally with 300 or 500 mg/kg twice daily for 6 days. Therapeutic treatment at 24 or 48 h post-infection in intranasally exposed, VEEV TrD-infected mice resulted in 90% and 40% survival, respectively, when dosed orally with EIDD-1931 at 500 mg/kg/day twice daily for 6 days [162]. These studies established the oral efficacy of EIDD-1931 against VEEV TrD in mice, providing a broader spectrum of activity for the compound and insights for further development.

### 4.5. Targeting Viral Replication through Alternative Modes of Action

**ML336** is a benzamidine-based small molecule that demonstrates potent antiviral activity in VEEV TC83-infected mice (Figure 11, Table 3) [163]. Several studies have come into view since 2017 that explored various aspects of ML336. This includes an alternative formulation of ML336, examination of the breadth of ML336 antiviral activity in vitro and in vivo, assessment of ML336-associated, resistance-conferring VEEV mutations, or survey of benzamidine structure–activity and structure–property relationships.

For instance, citing limitations in solubility and plasma stability, a study was undertaken to evaluate the use of lipid-coated mesoporous silica nanoparticles (LC-MSN) as a vehicle to deliver ML336 with anticipated increases in stability, solubility, tissue-targeting, and circulation time [164]. LC-MSNs contained about 20 μg ML336/mg LC-MSN and maintained colloidal stability for up to 4 days. In HeLa cells, ML336-loaded LC-MSNs inhibited VEEV TC83 replication up to a maximum of 6 log and without discernable cytotoxicity. In VEEV TC83-infected mice, ML336-loaded LC-MSNs resulted in a 10-fold reduction in brain viral load after 4 days compared to PBS-treated, infected mice [164], thus highlighting the potential of this delivery platform in the context of VEEV infection.

To better understand how ML336 exerts its antiviral effects, a more granular assessment of viral and host cell RNA synthesis was studied when ML336 treatment was applied [165]. Using nine structural analogs of ML336, it was shown that the most potent VEEV TC83 CPE assay inhibitors reduced viral RNA synthesis by the greatest amount. While ML336 did not appreciably affect host cell RNA synthesis, strand-specific qRT-PCR experiments showed that ML336 inhibited both positive and negative strand viral RNA synthesis. Further, ML336 inhibited the synthesis of both genomic and subgenomic viral RNA, and inhibition of all stages of viral RNA synthesis was dose dependent. These findings were also observed using the viral replicase enriched, membranous P15 fraction from VEEV-infected BHK cells. Collectively, these studies showed that benzamidine ML336 potently inhibited the synthesis of all VEEV RNA species through the intermediacy of the viral replication complex [165].

Previously, key in vitro mutations which rendered variant viruses inert to the effects of ML336 had been identified in both nsP2 (Y102C) and nsP4 (Q210K) [165,166]. In a follow-up study, the incidence and magnitude of resistance mutations was studied in nonhuman primate kidney epithelial cells and human astrocytes (Vero 76 and SVGA, respectively). The approach employed whole genome next-generation sequencing (NGS), thereby revealing single-nucleotide polymorphisms (SNPs) from passaged VEEV TC83 in the presence of the compound [167]. Notable outcomes included a common nsP4 Q210 mutation that dominated in both Vero 76 and SVGA cells. SNPs appeared more slowly in the SVGA cells, owing to their ability to produce type-1 IFNs. The major mutations were stable to additional passages in the absence of ML336 and maintained fitness. While RNA isolated from the brains of VEEV TC83-infected, ML336-treated mice were analyzed by NGS, depth of coverage was low, little to no overlap was observed between SNPs from in vivo and in vitro sources, and SNPs from mice were not ones known to confer resistance. A network analysis of the data showed that the microenvironment in these studies played a significant role in the evolution of mutations [167].

In a separate study, several analogs of ML336 were designed and synthesized to improve plasma stability and solubility without sacrificing VEEV potency [166]. The resulting benzamidine, **BDGR-4** (Figure 11, Table 3), differs structurally from ML336 with the inclusion of a 4-methoxy group on the *N*-amide-like portion of the scaffold which enhanced solubilization by 2.6-fold and resulted in 11.6% more of the parent compound remaining after liver microsome incubation. This structural change was integrated into subsequent analogs such as BDGR-5 and enantiomeric benzamidines **BDGR-69** and **BDGR-70** which also featured a strategically placed methyl substituent to potentially thwart undesirable amidine hydrolysis in plasma. While all analogs were similar in potency, ultimately BDGR-4 delivered the best boost in solubility while maintaining plasma stability and protein binding characteristics like that of ML336. Testing of BDGR-4 in CPE assays against WEEV California and EEEV FL93-939 showed EC_50_ values in the 102–150 nM range. Prophylactic administration of BDGR-4, BDGR-69, or BDGR-70 to mice challenged intranasally with VEEV TC83 showed that BDGR-4 provided the best protection, resulting in 67% survival in a 21-day study after dosing BDGR-4 at 2.5 mg/kg b.i.d. for 5 days [166]. In a head-to-head comparison between BDGR-4 and ML336, mice challenged subcutaneously with VEEV TrD after being treated with either compound resulted in 100% protection while all untreated mice died by day 7. Delay of treatment studies in VEEV TrD-infected mice showed that BDGR-4 provided full protection when dosing was initiated at 24 h post infection and 88% survival when dosing was started at 48 h post infection [166]. In vivo efficacy was also established for BDGR-4 against EEEV FL93-939 infection in mice where prophylactic treatment resulted in 90% survival. Viral titers in brain tissue were determined and changes in weight for in study animals were catalogued for these studies. BDGR-4 did not induce type-1 IFNs, indicating that the antiviral activity of BDGR-4 was not a result of activation of the host immune response. Identification of resistance mutations from in vitro studies revealed mutations in nsP4 (several) and nsP2 (one) whose emergence was BDGR-4 concentration dependent. The results of these collective studies extended the in vivo efficacy profile of ML336 and, in comparison to structural analogs, showed that BDGR-4 offered a significant solubility improvement over ML336 that enabled in vivo assessments in VEEV and EEEV infected mice. Those experiments highlighted BDGR-4 as a potent inhibitor of VEEV and EEEV that offers significant protection in lethal murine models of infection [166].

Dibenzylamine hit **compound 1**, discovered in a CPE-based high throughput screen, potently inhibited a VEEV TC83 CPE with an EC_90_ value of 0.89 µM, without cytotoxic liability up to a concentration of 30 µM, and reduced VEEV titer at a concentration of 10 µM by 7.49 log (Figure 11, Table 3) [168]. Nguyen and coworkers generated 24 analogs of compound 1 that surveyed three structural scaffold regions to retain or improve the antiviral activity while addressing limitations in solubility and microsomal stability. Modifications to the ring fused 1,4-dioxane moiety and alterations of the fluoromethoxybenzene component rendered analogs less potent than compound 1 in the CPE assay and did not improve microsomal stability for those compounds that were assessed. Some modest gains in CPE potency were observed for a benzylic gem-dimethyl substitution of the monomethyl benzylic linker; however, microsomal stability remained similarly compromised. Compound 1 was inactive against a panel of Old World alphaviruses but did inhibit VEEV replication at an early stage beyond viral entry, resulting in a blockage of VEEV RNA and protein production. Further refinement of microsomal stability and mechanism of action studies were proposed for this structural series [168].

**Table 3 viruses-15-00413-t003:** In vitro and in vivo anti-EEV data for direct acting antiviral small molecules.

Compound	Virus Strain	In Vitro Antiviral Data					In Vivo Antiviral Data		Ref.
		Cell Line	MOI	EC_50_ μM	CC_50_ μM	Titer Reduction (Concentration)	Mouse *Challenge*	Survival *Dose*	
compound 11	VEEV TC83	BE(2)M17	1.2	1.4	>25	-	*-*	*-*	[140]
	VEEV TC83	Neuro-2a	3.5	2.0	>25	-	*-*	*-*	[140]
	VEEV TrD	HeLa	0.1	1.6	>30	-	*-*	*-*	[140]
	VEEV TrD	Vero	0.01	2.4	30	-	*-*	*-*	[140]
compound 13	VEEV TC83	BE(2)M17	1.2	3.3	>25	-	*-*	*-*	[140]
	VEEV TrD	Neuro-2a	3.5	3.7	>25	-	*-*	*-*	[140]
SRI-33394	VEEV TC83	NHDF	1	0.77	>30	8.98 log (10 µM)	*-*	*-*	[141]
SRI-34329	VEEV TC83	NHDF	1	0.12	>50	5.96 log (10 µM)	*-*	*-*	[141]
tomatidine	VEEV TC83	U87MG	0.1	2.5	175	11-fold (10 µM)	*-*	*-*	[151]
	VEEV TrD	U87MG	0.1	-	175	364-fold (20 µM)	*-*	*-*	[151]
	EEEV FL93-939	U87MG	0.1	-	175	314-fold (20 µM)	*-*	*-*	[151]
Z-VEID-FMK	VEEV TC83	U87MG	0.1	0.5	>150	128-fold (10 µM)	*-*	*-*	[151]
	VEEV TrD	U87MG	0.1	-	175	887-fold (20 µM)	*-*	*-*	[151]
	EEEV FL93-939	U87MG	0.1	-	175	100-fold (20 µM)	*-*	*-*	[151]
citalopram HBr	VEEV TC83	U87MG	0.1	1	>150	87-fold (10 µM)	*-*	*-*	[151]
	VEEV TrD	U87MG	0.1	-	175	19-fold (20 µM)	*-*	*-*	[151]
	EEEV FL93-939	U87MG	0.1	-	175	17-fold (20 µM)	*-*	*-*	[151]
NHC or EIDD-1931	VEEV TC83	Vero	0.5	0.426	>200	2 log (1 µM) 4 log (2.5 µM)	*-*	*-*	[158]
	VEEV TrD	-	-	-	-	-	CD-1 *intranasal* *100 LD_50_*	90% ^a,b^ *300 mg/kg,* *BID, 6 d, PO*	[162]
	VEEV TrD						CD-1 *intranasal* *100 LD_50_*	90% ^a^ (+24 h PI) ^c^ *500 mg/kg,* *BID, 6 d, PO*	[162]
	VEEV TrD						CD-1 *intranasal* *100 LD_50_*	40% ^a^ (+48 h PI)^c^ *500 mg/kg,* *BID*, *6 d*, *PO*	[162]
ML336	VEEV TC83	Vero76	0.05	0.032	>50	7.0 log (5 µM)	C3H/HeN *intranasal* *10LD_50_*	71% ^b,d^ *5 mg/kg,* *BID, 4 d, IP*	[163]
	VEEV TrD	Vero76	0.05	0.04	>50	BLD (1 µM) BLD (0.5 µM)	BALB/c *SC challenge* *10LD_50_*	100% ^a,b^ 25 mg/kg, *BID, 8 d, IP*	[163] [166]
ML336/LC-MSN	VEEV TC83	Vero	0.1	-	-	6 log (2.5µg/mL)	C3H/HeN *intranasal* *10^8^ PFU*	reduced viral brain titer by 10-fold	[164]
BDGR-4	VEEV TC83	Vero76	0.05	0.047	>50	7.0 log (5 µM)	C3H/HeN *SC challenge* 1 × 10^7^ PFU	100% ^b,e^ *5 mg/kg,* *BID, 8 d, IP*	[166]
	VEEV TrD	-	-	-	-	-	BALB/c *SC challenge* *10LD_50_*	100% ^a,b^ *25 mg/kg,* *BID, 8 d, IP*	[166]
	VEEV TrD	-	-	-	-	-	BALB/c *SC challenge* *10LD_50_*	100% ^a^ (+24 h PI) ^c^ *25 mg/kg,* *BID, 8 d, IP*	[166]
	VEEV TrD	-	-	-	-	-	BALB/c *SC challenge* *10LD_50_*	88% ^a^ (+48 h PI) ^c^ *25 mg/kg,* *BID, 8 d, IP*	[166]
	WEEV California	Vero76	0.05	0.102	>50	>6.2 log (5 µM)	*-*	*-*	[166]
	EEEV FL93-939	Vero76	0.05	0.149	>50	-	C57BL/6 *SC challenge* *10^4.3^ CCID_50_*	90% ^a,b^ *50 mg/kg,* *BID, 8 d, IP*	[166]
BDGR-69	VEEV TC83	Vero76	0.05	0.028	>50	7.2 log (5 µM)	C3H/HeN *intranasal* 1 × 10^7^ PFU	50% ^a,b^ *25 mg/kg,* *BID, 5 d, IP*	[166]
BDGR-70	VEEV TC83	Vero76	0.05	0.025	>50	7.2 log (5 µM)	C3H/HeN *intranasal* 1 × 10^7^ PFU	67% ^a,b^ *25 mg/kg,* *BID, 5 d, IP*	[166]
	VEEV TC83	Vero76	0.05	0.025	>50	7.2 log (5 µM)	C3H/HeN *intranasal* 1 × 10^7^ PFU	100% ^b,e^ *2.5 mg/kg,* *BID, 8 d, IP*	[166]
compound 1	VEEV TC83	THFF	1	0.89 ^f^	>30	7.5 log (10 µM)	-	-	[168]

Only compounds with in vitro or in vivo data for any of the EEVs is tabulated; EC_50_, effective concentration required to inhibit 50% of the treated cell population; CC_50_, cytotoxic concentration that results in death of 50% of the treated cell population; BID, twice daily; PO, oral administration; SC, subcutaneous administration; IP, intraperitoneal administration; BLD, below limits of detection; ^a^ compared to 0% survival in untreated controls; ^b^ prophylactic study with dosing initiated two hours prior to viral challenge; ^c^ delay of treatment study with dosing initiated at the indicated time point post-infection (PI); ^d^ survival in untreated control group was 14%. ^e^ survival in untreated control group was 17%; ^f^ EC_90_ value.

## 5. Additional Small Molecule Inhibitors of Encephalitic Alphaviruses

Compounds were published with activity against VEEV, WEEV, and/or EEEV that are not discussed in the preceding sections. Generally, these compounds have yet to be subjected to in-depth mechanistic studies or divergent mechanisms are suggested to account for the observed activity, thereby complicating a straightforward categorization. In some cases, the compounds were identified from studies centered on other viruses which reported anti-EEV activity as part of the antiviral spectrum. Nonetheless, these compounds are important to track in the development pipeline as they may hold insights into unique or broader approaches to preventing or treating encephalitic alphavirus infection.

The synthetic chemistry that led to the discovery and development of anti-VEEV benzamidines ML336 and BDGR-4 [163,166] was recently modified, leading to the formation of a new class of benzodiazepinones [169]. A structural model overlaying the benzodiazepinone framework with BDGR-4 showed remarkable alignment. As such, a library of 17 benzodiazepinones was prepared bearing substituents known to impart anti-VEEV activity on the benzamidine core. The resulting benzodiazepinones were assessed in CPE and titer reduction assays using VEEV INH9813 or EEEV V105 in Vero 76 cells. The most potent compounds (EC_50_ = 27–48 nM for VEEV and EEEV) featured a C8 nitro group, though replacement with a nitrile moiety was possible in combination with other structural changes without significant potency loss. At a concentration of 5 µM, VEEV and EEEV titers were reduced by >5 log for many examples. **Compounds 7a**, **7b**, **7o** and the separated enantiomers of 7o were evaluated for VEEV and EEEV yield reduction in human brain primary neuronal cells (Figure 12, Table 4). At a concentration of 5 µM, viral titers were reduced to the limits of assay detection. Compound 7o and its individual isomers, assessed at 1 µM, were not significantly different from each other and resulted in at least a 2 log reduction in VEEV and EEEV titers. Aqueous solubility was modest for several compounds, and microsomal stability was marginal for compounds 7a and 7b (t_1/2_ = 10 min) but was improved for 7o (t_1/2_ = 45 min). The need for mechanism of action insights and structural modifications to improve microsomal stability were discussed to advance this new chemotype with potent antiviral cell activity against both VEEV and EEEV [169].

In a study aimed at developing inhibitors of CHIKV, select compounds that had been originally tested against VEEV were evaluated against CHIKV for new medicinal chemistry opportunities [170]. **Compound 1a** (Figure 12, Table 4) modestly inhibited VEEV TC83 (EC_50_ = 13.2 µM, Vero cells) [163], but 1a demonstrated improved CHIKV potency and was selected as an optimization point, leading to the development of **compound 8q**. In normal human dermal fibroblast (NHDF) cells, compound 8q inhibited several alphaviruses, including VEEV TC83, with an EC_90_ value of 0.40 µM without cytotoxicity at the highest concentration tested (CC_50_ > 30 µM). At a concentration of 10 µM, VEEV viral titer was reduced by 3.1 log. Pharmacokinetic parameters for compound 8q were determined in mice, which may inform potential VEEV-centered studies. Resistant mutations in CHIKV implicated involvement of the nsP3 macrodomain in the compound’s mechanism of action. Compound 8q also inhibited human dihydroorotate dehydrogenase (IC_50_ = 0.31 µM), a host target whose inhibition has been associated with in vitro antiviral activity due to the role of the enzyme in nucleotide biosynthesis [170,171,172]. In sum, these efforts may help guide additional work with this scaffold on encephalitic alphaviruses.

In a separate CHIKV-focused study, **lycorine** (Figure 12, Table 4) was assessed for activity in cells infected with various alphaviruses [173]. Lycorine is an alkaloid derived from plants that has been studied for a range of pharmacological effects, including antiviral activity [173,174] (see Appendix A Figure A1 for structural note). The compound was found to inhibit CHIKV infection in various cell types with submicromolar IC_50_ values and without apparent cytotoxicity. Lycorine was tested against a panel of alphaviruses, including VEEV, which revealed an IC_50_ of 0.31 µM. Several mechanisms have been proposed to account for the observed effects of lycorine on other viruses [175,176,177,178] and in this study, the time of addition studies and subsequent experiments with an nsP4-inactivated viral mutant suggest that lycorine inhibited CHIKV replication by interrupting viral RNA translation [162].

**Homoseongomycin** is a marine natural product that was found during a high throughput screen [179] aimed at identifying compounds with activity against VEEV (Figure 12, Table 4). In Vero cells, homoseongomycin inhibited VEEV with EC_50_ values of 8.6 µM (TC83-luc assay) and 9.1 µM (ZPC738 strain). At a compound concentration of 50 µM, viral titers were reduced by 8 log (TC83, Vero cells) and 4 log (ZPC738, U87MG cells) with no observable toxicity up to 50 µM in either cell line. A nano luciferase reporter virus of EEEV FL93-939 was also inhibited by homoseongomycin (EC_50_ = 1.2 µM). Time of addition studies showed that homoseongomycin significantly inhibited VEEV entry as well as later stages of viral infection. The intermediacy of host factors in the observed antiviral effects of homoseongomycin is undetermined at this time but these experiments and assessments in animal models are proposed [179].

Triazavirin (riamilovir) is a non-nucleoside triazolotriazine-based analog that has been investigated for its antiviral activity against tick-borne encephalitis and influenza [180,181]. To assess if derivatives of triazavirin would be effective in VEEV and EEEV cell culture and in infected mice, analogs were generated as sodium salts that incorporated a nitrile group in place of the nitro group of the parent structure, and then modifications were made to the triazine thiomethyl substituent or in the triazine ring itself [181]. At a concentration of 100 µg/mL, a 1.9 and 2.6 log reduction in VEEV and EEEV titers, respectively, was observed for **compound 4b** (Figure 12, Table 4). In VEEV strain 230-infected mice (parental challenge), compound 4 was dosed orally at 50 or 100 mg/kg at 2 h post infection and continued for 5 days, resulting in 60% and 80% survival, respectively. Oral dosing initiated at 24 h post infection in VEEV-infected mice and continued for 4 days resulted in 40% and 70% survival at 50 or 100 mg/kg doses, respectively. Similar protocols employed for EEEV strain 463 efficacy studies showed 50% and 70% survival in mice dosed with 50 or 100 mg/kg/d at 2 h post-infection, respectively. For mice dosed at 24 h post infection, 40% and 70% survival were observed with 50 or 100 mg/kg/d of compound 4b, respectively. Mouse toxicology experiments did not reveal behavioral or morphological changes in mice up to 28 days with daily injections of 375 mg/kg [181].

A series of indole 2-carboxamides, represented by **compound 2** (Figure 12, Table 4), was described previously [182] with activity against WEEV in a replicon assay (IC_50_ = 0.5 µM, CC_50_ = 65 µM); however, the precise mechanism of action regarding the target and how the compounds may bind was unknown. Conformationally restricted indole 2-carboxamides were synthesized to improve potency against WEEV and define a pharmacophoric model [183]. Though significant gains in potency were not achieved compared to compound 2, **dihydroindene 12** showed comparable potency against WEEV without notable cytotoxicity (IC_50_ = 0.53 µM, CC_50_ > 100 µM). Scaffold rigidification and SAR, with computational analysis, refined a pharmacophoric model for this chemotype [183].

A collection of quinazolines and quinolines featuring a 4-aminoaryl group was designed based on structural similarities between the anti-DENV 4-aminoquinazoline, erlotinib, and a 4-aminoquinoline that demonstrated antiviral activity against both DENV and VEEV [184]. Following a systematic SAR effort, **quinoline 27** and **quinazoline 54** (Figure 12, Table 4) showed activity in human U87MG astrocytes infected with VEEV TC83 (EC_50_ = 0.50–0.60 µM, CC_50_ > 10 µM). In a separate study, structurally related compounds that more deeply assessed substitutions of the 4-aminoaryl substituent were assessed against these viruses [185], resulting in several analogs with single digit TC83 micromolar activity.

**Table 4 viruses-15-00413-t004:** In vitro and in vivo anti-EEV data for antiviral small molecules.

Compound	Virus Strain	In Vitro Antiviral Data					In Vivo Antiviral Data		Ref.
		Cell Line	MOI	EC_50_ μM	CC_50_ μM	Titer Reduction (Concentration)	Mouse *Challenge*	Survival *Dose*	
compound 7a	VEEV INH9813	Vero76	0.05	0.041	>30	5.5 log (5 µM)	-	-	[169]
	EEEV V105	Vero76	0.05	0.033	>30	7.9 log (5 µM)	-	-	[169]
compound 7o	VEEV INH9813	Vero76	0.05	0.24	23.3	5.2 log (5 µM)	-	-	[169]
	EEEV V105	Vero76	0.05	0.16	23.3	6.9 log (5 µM)	-	-	[169]
compound 1a	VEEV TC83	NHDF	1	2.1 ^c^	>30	-	-	-	[170]
compound 8q	VEEV TC83	NHDF	1	0.4 ^c^	>30	3.1 log (10 µM)	-	-	[170]
lycorine	VEEV TC83	Vero	0.01	0.31	>10	-	-	-	[173]
homoseon- gomycin	VEEV TC83-luc	Vero	0.1	8.6	>50	8 log (50 µM)	-	-	[179]
	VEEV ZPC738-luc	U87MG	0.1	9.1 ^a^	>50	4 log (50 µM)	-	-	[179]
	EEEV FL93-939-luc	Vero	0.1	1.2	>50	-	-	-	[179]
compound 4b	VEEV 230	PEKC	0.02	-	-	1.9 log PFU (100 µg/mL)	outbred albino *SC challenge* *10LD_50_*	60% ^b^ (+2 h PI) *50 mg/kg,* *QD, 5 d, PO*	[181]
	VEEV 230	PEKC	0.02	-	-	1.9 log PFU (100 µg/mL)	outbred albino *SC challenge* *10LD_50_*	80% ^b^ (+24 h PI) *100 mg/kg,* *QD, 5 d, PO*	[181]
	EEEV 463	PEKC	0.02	-	-	2.6 log PFU (100 µg/mL)	outbred albino *SC challenge* *10LD_50_*	50% ^b^ (+2 h PI) *50 mg/kg,* *QD, 5 d, PO*	[181]
	EEEV 463	PEKC	0.02	-	-	2.6 log PFU (100 µg/mL)	outbred albino *SC challenge* *10LD_50_*	70% ^b^ (+24 h PI) *100 mg/kg,* *QD, 5 d, PO*	[181]
compound 2	WEEV Cba-87	BSR-Z7/C3	0.1	0.53	65	-	-	-	[183]
dihydro- indene 12	WEEV Cba-87	BSR-Z7/C3	0.1	0.53	>100	-	-	-	[183]
quinoline 27	VEEV TC83	U87MG	0.1	0.50	>10	-	-	-	[184]
quinazoline 54	VEEV TC83	U87MG	0.1	0.60	>10	-	-	-	[184]

Only compounds with in vitro or in vivo data for any of the EEVs is tabulated; EC_50_, effective concentration required to inhibit 50% of the treated cell population; CC_50_, cytotoxic concentration that results in death of 50% of the treated cell population; PO, oral administration; SC, subcutaneous administration; ^a^ In Vero cells; ^b^ delay of treatment study with dosing initiated at the indicated time point post-infection (PI); ^c^ EC_90_ value.

## 6. Discussion

Alphavirus infection is a serious public health concern due to a constellation of factors. For instance, the absence of approved, effective, and safe prophylactic or therapeutic modalities creates vulnerabilities for which we are unprepared. This is exacerbated by the unpredictable nature of outbreaks and global climate changes that expand the geographic regions affected by mosquito-borne transmission. In addition, VEEV, WEEV and EEEV are classified by the National Institute of Allergy and Infectious Diseases (NIAID) as category B pathogens due to their biowarfare potential via aerosolization and deliberate distribution [186]. Further, human acquired infections caused by these viruses have been shown to place a substantial physical and economic burden on survivors [187,188]. Taken together, the need for translational research and outcomes in this area are underscored.

Herein, small molecules affecting EEVs were highlighted that have been discovered or advanced in development since a thorough review was done in 2017 [34]. In this report, assessment of compounds against specific targets, determination of cellular antiviral activity and spectrum, exploration of SAR and establishment of ADME, pharmacokinetic and toxicologic parameters, and examination of formulation were showcased. Additionally, several compounds were evaluated in mouse infection models for VEEV (8 compounds, 5 distinct chemotypes), WEEV (1 compound/chemotype), and EEEV (2 compounds/chemotypes), revealing significant survival outcomes in mice across these viruses. Collectively, these studies provide benchmarks and granularity on mechanisms of action that may be useful in the optimization and study of future analogs and chemotypes.

As new compounds are identified and existing antivirals are advanced for drug development purposes, it is important to define the target product profile (TPP). The TPP provides a roadmap to the experiments and data that will be required to demonstrate safety and efficacy and accounts for pipeline challenges that need to be addressed. Ideally, an EEV antiviral is orally administered, quickly distributed to the brain and other tissues with once-a-day dosing, efficacious and safe for the broadest patient population, inhibits all three EEVs in prophylactic and therapeutic scenarios and is independent of route of exposure, avoids the development of resistance, and is shelf stable without a cold chain requirement. To achieve these goals, long range planning is essential. For instance, the choice of cell lines, viral strain alignment from in vitro to in vivo assessment, the readiness of animal models that recapitulate human disease, biomarker identification, and even the outlook for clinical trial patient recruitment should be examined, as these factors may require adjustment of the development plan and experiments needed along the way.

Considerations for early-stage compounds include an understanding of mechanism, demonstration of antiviral activity and selectivity indices in relevant cell lines, and with wild-type strains of alphaviruses that will likely be used in downstream efficacy models. Establishment of structure–activity and structure–property relationships are important to indicate that the scaffold can tolerate structural modifications with improvement in multiple profile parameters. Physiochemical properties should be evaluated once compound hits are validated, as solubility and microsomal stability are critical guideposts for optimization. Brain exposure is also an important aspect of the compound profile given the need to penetrate the blood brain barrier (BBB) to inhibit EEV replication in neuronal cells. In vitro assays are available to assess BBB permeability and can be useful in the down-selection of compounds for advanced in vivo studies. Tiered ADME studies assist in compound prioritization for PK and efficacy studies, while also pinpointing potential liabilities that may be remedied through medicinal chemistry efforts. PK studies determine plasma and tissue exposure as a function of dose and route of administration and help guide dosing in animal models. Demonstration of efficacy and safety in validated animal models with viral strains that are clinically relevant are also critical milestones.

Drug discovery and development activities centered on EEVs are impeded by unique challenges, in addition to the traditional drug pipeline bottlenecks, risks, triage or failure observed during hit-to-lead, lead advancement, and preclinical studies. While very early activities may avoid the use of highly pathogenic viruses or select agents, at some point, specialized biosafety facilities, regulatory oversight, and trained personnel are required to execute studies with relevant virus strains. VEEV and EEEV are categorized as select agents, except for the attenuated VEEV TC83 strain that can be used at a BSL2 level. Therefore, highly pathogenic, wild-type strains of VEEV, WEEV, and EEEV are restricted to use within a limited number of BSL3 facilities. Additionally, our understanding of alphavirus pathogenesis and the intermediacy of host proteins continues to grow, hampered in part by the absence of X-ray crystal structures of some targets such as the nsP4 viral polymerase that would help guide compound optimization. Priming the pipeline with high-quality, drug-like chemical matter from the start is also imperative to enhance the success of translation. Data from validated animal models of EEV infection that strongly mirror the pathogenesis and hallmarks of the human disease and provide clear clinical endpoints are also critical. This point is especially poignant due to the applicability of the animal rule [189,190], as patient recruitment for an EEV human clinical efficacy study is more difficult given the unpredictable nature of EEV outbreaks and coincident infections that obscure straightforward diagnoses of encephalitic alphavirus infections.

Despite these hurdles, the outlook for finding small molecule derived drugs for EEVs is promising. Technological innovation in cryo-EM and genetic and analytical tools may enable examination of compound-target interactions that to date have been challenging to study. Compounds in the EEV pipeline have advanced further and with greater characterization than in previous years, as exemplified by EIDD-1931, which progressed within the EEV pipeline until a strategic pivot to SARS-CoV-2 was implemented, resulting in the emergency use authorization of Molnupiravir [191]. In fact, the global impact of COVID-19 has highlighted a gap in the availability and development of anti-infective agents for viruses that pose a significant threat or have pandemic potential [192,193]. Consequently, funding initiatives have been established to support efforts in this area which includes the *Togaviridae* EEVs, VEEV and EEEV, as viruses of concern. In addition to EIDD-1931, the benzamidine BDGR-4 is notable due to its in vitro activity against all three wild-type EEVs and demonstrated in vivo efficacy against VEEV and EEEV. These and other compounds that advance into higher species for efficacy and safety may serve as informative benchmarks by which better antivirals will be designed. Ultimately, this momentum in the field enhances the likelihood that safe and effective small molecule-based drugs can be developed to address gaps in the EEV pipeline.

## Figures and Tables

**Figure 1 viruses-15-00413-f001:**
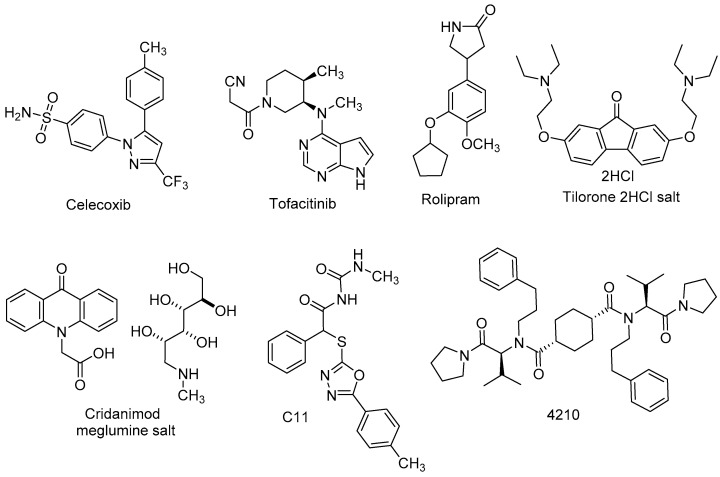
Chemical structures of compounds that target the host interferon or anti-inflammatory response.

**Figure 2 viruses-15-00413-f002:**
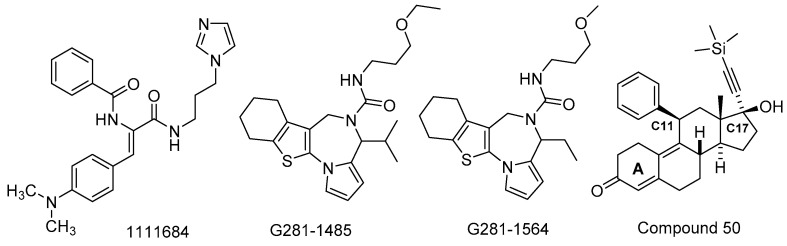
Chemical structures of compounds that inhibit nuclear protein transport.

**Figure 3 viruses-15-00413-f003:**
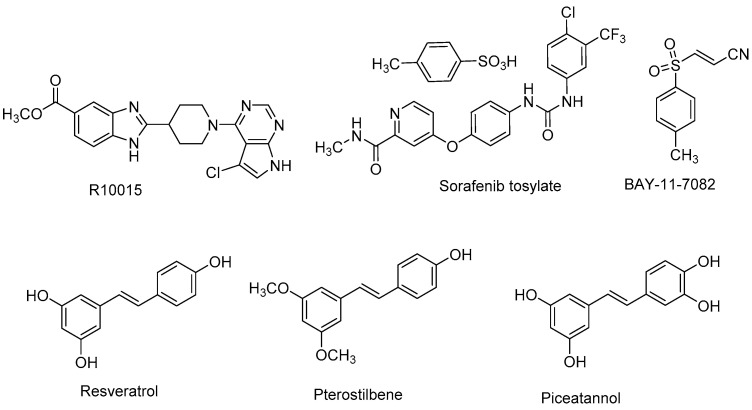
Chemical structures of compounds that alter host protein phosphorylation.

**Figure 4 viruses-15-00413-f004:**
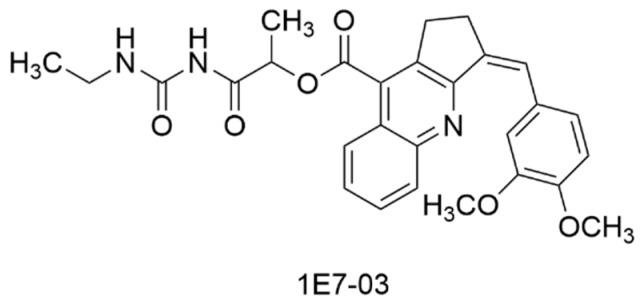
Chemical structure of 1E7-03, a protein phosphatase 1α binding motif inhibitor.

**Figure 5 viruses-15-00413-f005:**
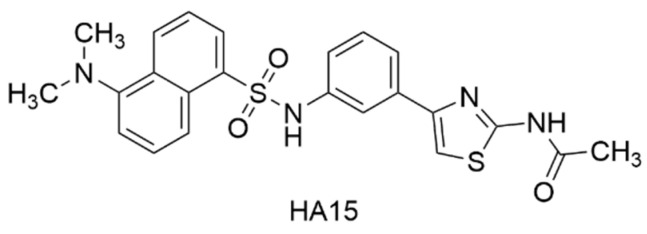
Chemical structure of host protein GRP78 inhibitor, HA15.

**Figure 7 viruses-15-00413-f007:**
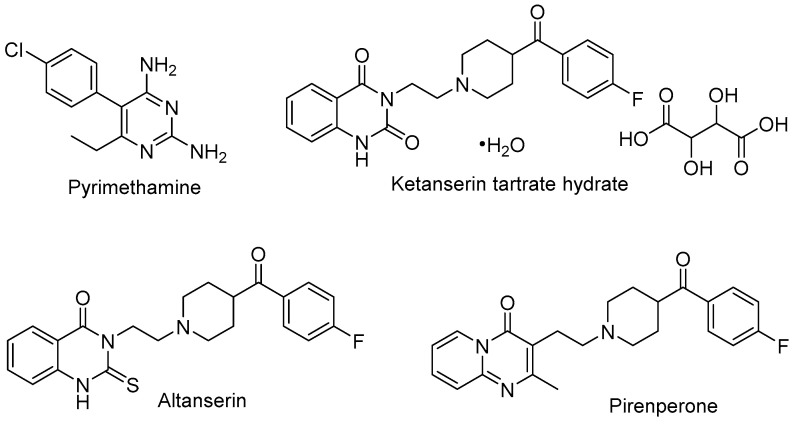
Chemical structures of compounds that target VEEV nsP1.

**Figure 8 viruses-15-00413-f008:**
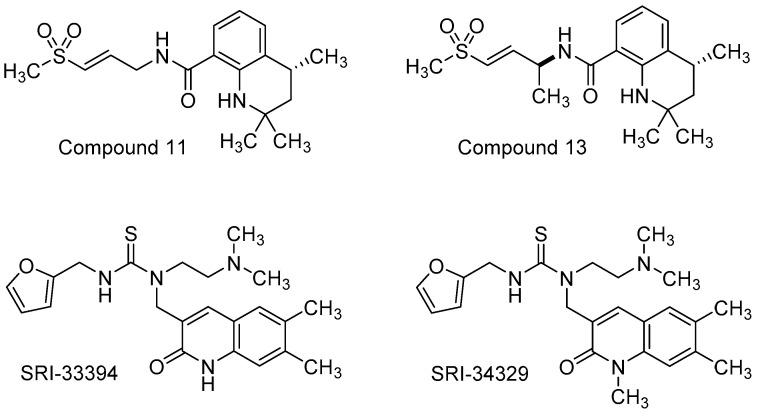
Chemical structures of compounds that target nsP2.

**Figure 9 viruses-15-00413-f009:**
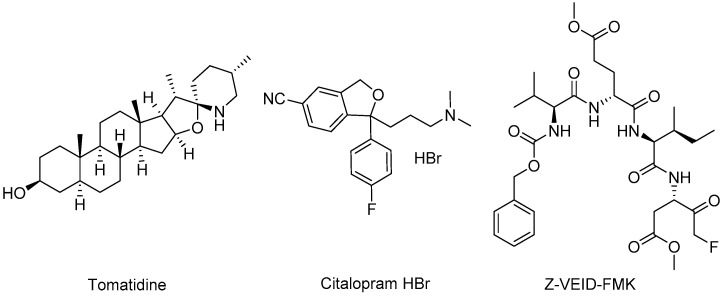
Chemical structures of compounds that target nsP3.

**Figure 10 viruses-15-00413-f010:**
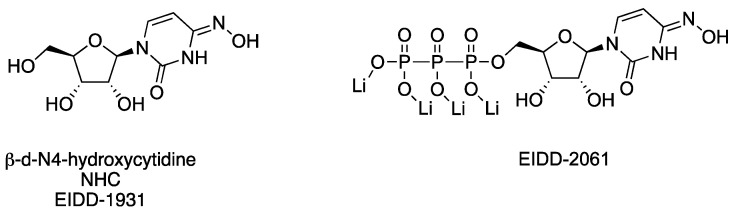
Chemical structures of compounds that target nsP4.

**Figure 11 viruses-15-00413-f011:**
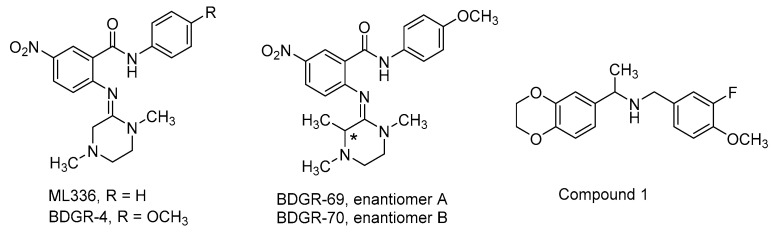
Chemical structures for compounds that that target viral replication.

**Figure 12 viruses-15-00413-f012:**
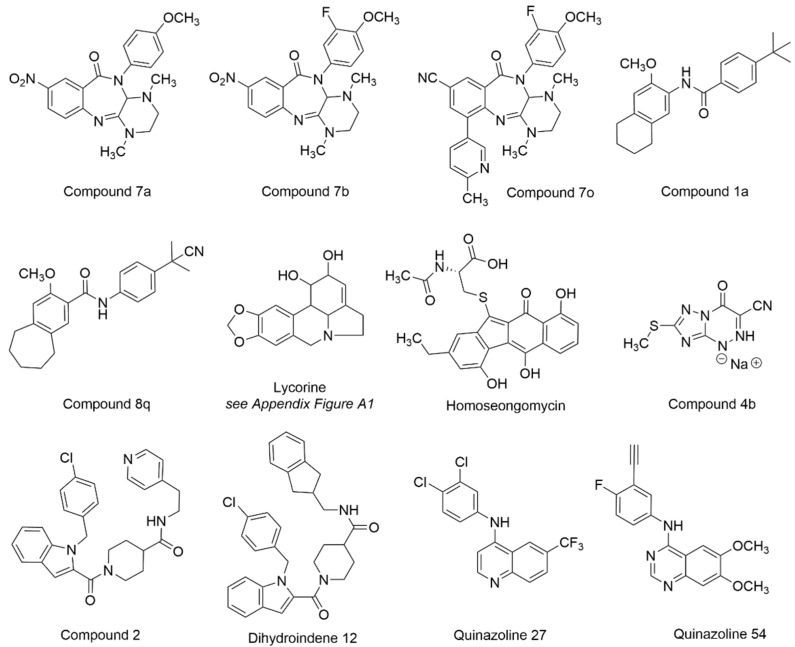
Chemical structures for compounds intervening in the EEV life cycle.

**Table 2 viruses-15-00413-t002:** In vitro data for compounds that target VEEV non-structural protein 1.

Compound	Virus	In Vitro Activity		Ref.
		GT Reaction, IC_50_, μM	MTase Reaction, % Inhibition at 50 µM	
pyrimethamine	VEEV P676-nsP1 Protein	2.7 μM	73.5%	[124]
ketanserin	VEEV P676-nsP1 Protein	14.6 μM	79.3%	[124]
pirenpirone	VEEV P676-nsP1 Protein	39.6 μM	44.6%	[124]
altanserin	VEEV P676-nsP1 Protein	9.3 μM	-	[124]

## Data Availability

Not applicable.

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
