# Peer review of "Advances in the Development of Small Molecule Antivirals against Equine Encephalitic Viruses"

_viruses, 2023, doi:10.3390/v15020413_

Round 1
Reviewer 1 Report
The authors provide a comprehensive and well written review of EEV antivirals. Only minor edits are suggested.
Line 53: Please provide further details about the truncated TC83 vaccine.
Line 91-92: Six proteins are produced from the subgenomic RNA. 6K and TF are two separate proteins produced through frameshifting. Please clarify this.
Line 194: Please clarify what is meant by “cTC83/TrD infected mice”. As written, it appears that mice may be infected with both strains.
Line 212 – Please be consistent in how U-87 MG cells are written.
Table 3 – Table 3 is broken up into 3 sections. However, it appears that they could be merged together or at least only split into 2 sections.
Author Response
We thank the reviewer for the comments and suggestions. Please find the following responses and edits, detailed below.
Rev 1, item 1: Line 53: Please provide further details about the truncated TC83 vaccine.
Author response: This was incorrectly written and has been revised to read, “Inactivated vaccines are available to protect equids, and either an inactivated or live, attenuated VEE TC83 vaccine has been provided for select laboratory and military personnel risked exposure to these viruses; however, these vaccines are not suitable for use by the general public due to questions of efficacy and safety.” – lines 53-56
____________________________________________________________________________________
Rev 1, item 2: Line 91-92: Six proteins are produced from the subgenomic RNA. 6K and TF are two separate proteins produced through frameshifting. Please clarify this.
Author response: This portion was revised to now read as “A single, positive-sense RNA strand contains the alphavirus genome. Within two open reading frames are encoded six structural proteins and four non-structural proteins (nsPs). A capsid protein (CP), envelope proteins (E1, E2, E3), a viroporin 6K protein, and a virulence factor known as the transframe (TF) protein [35] represent the structural proteins needed for cell entry, host defense mitigation, virulence, encapsidation and budding events [36, 37].” A reference was added to support the description of 6K and TF.
_____________________________________________________________________________________
Rev 1, item 3: Line 194: Please clarify what is meant by “cTC83/TrD infected mice”. As written, it appears that mice may be infected with both strains.
Author response: This was the molecular clone and designation defined by the authors of the study. We have added a clarification in our text to characterize this designation: “The cTC83/TrD virus, so named because it contains two nucleotide substitutions from the TC83 genome, resulting in increased homology and virulence as that of wild-type Trinidad donkey (TrD) strain, was confirmed to induce a lethal cytopathic effect in BHK cells [63]."
_____________________________________________________________________________________
Rev 1, item 4: Line 212 – Please be consistent in how U-87 MG cells are written.
Author response: Fixed and normalized throughout.
_____________________________________________________________________________________
Rev 1, item 5: Table 3 – Table 3 is broken up into 3 sections. However, it appears that they could be merged together or at least only split into 2 sections.
Author response: This was an unintended artifact of the submission process that we also noted when the draft came back from review that several tables suffered this effect. We reformatted the tables to minimize continued tables when possible.
Reviewer 2 Report
The review discusses various small molecule inhibitors of potential use against equine encephalitic alpha-viruses. In this regard the study is important in consolidating the knowledge in this area. Especially in view of potential future outbreaks/pandemics involving an alpha-virus. Such information would provide readily available choices for possible mitigating interventions. As such the review contributes to knowledge in the field and is commendable.
· Lines 15-17, rewrite sentence for grammatical flow
· Line 493, line 515 and elsewhere in the text; no need to quote year of work done, the reader can find that information in the cited paper
· 518, spelling correction
· The authors in their discussion and conclusion should point out from all the compounds considered which one are most promising to be advanced for use.
Author Response
We thank the reviewer for the comments and suggestions. Please find the following responses and edits, detailed below.
Rev 2, item 1: Lines 15-17, rewrite sentence for grammatical flow
Author response: The sentence in questions was revised: “Herein, small molecule inhibitors of VEEV, WEEV, and EEEV are reviewed that have been identified or advanced in the last five years since a comprehensive review was last performed.”
_____________________________________________________________________________________
Rev 2, item 2: Line 493, line 515 and elsewhere in the text; no need to quote year of work done, the reader can find that information in the cited paper
Author response: We took out several of these instances, including the one referred to specifically, on the advice of the reviewer.
_____________________________________________________________________________________
Rev 2, item 3: 518, spelling correction
Author response: ‘starting’ was fixed.
_____________________________________________________________________________________
Rev 2, item 4: The authors in their discussion and conclusion should point out from all the compounds considered which one are most promising to be advanced for use.
Author response: The last paragraph has been revised to provide a more complete perspective on the compounds discussed therein.
Reviewer 3 Report
The manuscript of Ogorek and Golden represents a comprehensive review of small molecule antivirals tested to date against Equine Encephalitis viruses. This is a very well written review with a deep analysis of the topic. It was interesting to read the review, although I believe the article could’ve been shorter. Besides of that I have no other suggestions to this manuscript.
Author Response
We thank the reviewer for the comments and suggestions.
REV 3 “The manuscript of Ogorek and Golden represents a comprehensive review of small molecule antivirals tested to date against Equine Encephalitis viruses. This is a very well written review with a deep analysis of the topic. It was interesting to read the review, although I believe the article could’ve been shorter. Besides of that I have no other suggestions to this manuscript.”
Author response: We have gone through and made edits to cut down extraneous content and revise to shorten where possible.
Reviewer 4 Report
Excellent review. One of the best reviews about small molecule antivirals I have read. Couple of suggestions.
1) I understand there is not room in the current tables, but I think it would add extra value to the review if there was a table that showed the molecules in which resistance/escape mutants were identified and where and what mutations those were
2) Truncated VEEV TC83 was not used to provide select laboratory and military personnel. As part of the Special Immunizations Program (SIP), select laboratory and military personnel were administered full length TC83 or C83 (inactivated full length TC83).
3) In section 2 line 103- are the alternative entry mechanisms fall along differences in new world alphaviruses vs old world alphaviruses. It appears by how the sentence is written that it does, but further clarification by be helpful.
4) Line 185-187: Could the confounding studies be due to differing ability of the cells to induce type I IFN? PBMCs respond rapidly to type I IFN and induce large amounts of type I IFN while fibroblast dont respond as rapidly to type I IFN and induce less. Additionally, many fibroblast cell lines are deficient in part of the type I IFN pathway. For example, Vero cells respond to type I IFN but do not induce it. I would add a little context to this area to say why there could be conflicting studies.
5) In the tables, it would be helpful to add challenge dose under the mouse model section.
6) Line 473- Should be At not Aa
7) Lines 765-788: A lot of time is spent talking about CHIKV in this section. Since this is about VEEV, EEEV, and WEEV I would cut this down to a couple of sentences especially since there has been a lot of small molecule work done with CHIKV and that is not reflected in this review.
8) In line 881 it is mentioned that it is important to pick relevant cell lines, I think that relevant virus strains should be added to this. There are many compounds that only test against TC83 which is a vaccine strain that is attenuated and can give false positives and failures when moving forward to WT strains. Many use TC83 because it is a BSL-2 strain so more labs can use this for initial test, but many times most of the work is done with TC83 and then one little figure at the end saying, oh and in a plaques assay it also works against VEEV Trd. Especially when looking at host directed responses, TC83 produces a vastly different inflammation profile than VEEV TrD or other WT strains.
9) Line 893-894 is a really important statement that should be expanded upon by at least a couple of sentences. The biggest gap in the small molecule antiviral field is the lack of compounds that make it into in vivo testing. There is lots of in vitro work but very little in vivo work, especially with WT strains. Apart of the expanding of this could be what are some of the difficulties that lead to less in vivo testing, lack of facilities, lack of ability to work with select agents, etc.
Author Response
We thank the reviewer for the comments and suggestions. Please find the following responses and edits, detailed below.
Rev 4, item 1: I understand there is not room in the current tables, but I think it would add extra value to the review if there was a table that showed the molecules in which resistance/escape mutants were identified and where and what mutations those were
Author response: Unfortunately, we hadn’t culled that data as we went through the review of the articles but rather, we described key mutations induced in vitro with increasing compound concentrations in the text as compounds were described. We added obvious ones that we had left out for a few compounds based on this request, but to go back and tabulate concentration dependent listings of mutations is beyond what we have space for in this review.
Rev 4, item 2: Truncated VEEV TC83 was not used to provide select laboratory and military personnel. As part of the Special Immunizations Program (SIP), select laboratory and military personnel were administered full length TC83 or C83 (inactivated full length TC83).
Author response: This has been revised to read, “Inactivated vaccines are available to protect equids, and either an inactivated or live, attenuated VEE TC83 vaccine has been provided for select laboratory and military personnel risked exposure to these viruses; however, these vaccines are not suitable for use by the general public due to questions of efficacy and safety.” – lines 53-56
Rev 4, item 3: In section 2 line 103- are the alternative entry mechanisms fall along differences in new world alphaviruses vs old world alphaviruses. It appears by how the sentence is written that it does, but further clarification by be helpful.
Author response: The sentence has been rewritten as it was not intended to suggest that there was different mechanisms according to Old vs New World alphaviruses. Rather, we wanted to acknowledge studies that showed exceptions to only clathrin-mediated endocytosis. These studies are limited to only one virus and/or certain conditions, such as pH or cell type. Therefore, it is challenging to broadly characterize the alphaviruses until further studies are done. We have revised as follows:
Generally, alphaviruses utilize clathrin-mediated endocytosis to infect host cells [44], though caveolae-mediated endocytosis [45] of Mayaro virus (MAYV), micropinocytosis [46] of CHIKV in human muscle cells, and pH-dependent pore formation with SINV [47] has also been described [48].”
Rev 4, item 4: Line 185-187: Could the confounding studies be due to differing ability of the cells to induce type I IFN? PBMCs respond rapidly to type I IFN and induce large amounts of type I IFN while fibroblast dont respond as rapidly to type I IFN and induce less. Additionally, many fibroblast cell lines are deficient in part of the type I IFN pathway. For example, Vero cells respond to type I IFN but do not induce it. I would add a little context to this area to say why there could be conflicting studies.
Author response: Agreed, differences in IFN response among cell types is documented and may very well account for some of the observed differences. Our use of the word “confoundedly” likely suggested that there were no hypotheses to account for observed discrepancies, and as such, we revised this sentence to better reflect the intent. While it could be that the variability of IFN induction across cell types is responsible for some of the observed disconnects, it probably doesn’t account for all of them. There is also discussion in the literature on lack of engagement with the hSTING pathway and multiple, ambiguous mechanistic possibilities for these particular compounds that may cloud interpretation. We had included that point at the end of the paragraph. A broader discussion of all of this is beyond the scope of this review but we have revised the section pointed out as follows:
“Tilorone and cridanimod (Figure 1, Table 1) are among the earliest small molecules identified that potently induce IFN production in murine macrophages and mice through the intermediacy of the mSTING pathway [58]. While studies in human cells such as PBMCs, fibroblasts, and HEK293T cells, which notably have variable responses to IFN induction, revealed that these compounds did not similarly engage the hSTING pathway or produce a strong IFN response in human patients [58, 59], literature documentation of in vitro and clinical antiviral efficacy abounds, and the compounds have been used clinically as antivirals in humans in various countries [60-62].”
Rev 4, item 5: In the tables, it would be helpful to add challenge dose under the mouse model section.
Author response: Agreed. This data has been added.
Rev 4, item 6: Line 473- Should be At not Aa
Author response: Fixed.
Rev 4, item 7: Lines 765-788: A lot of time is spent talking about CHIKV in this section. Since this is about VEEV, EEEV, and WEEV I would cut this down to a couple of sentences especially since there has been a lot of small molecule work done with CHIKV and that is not reflected in this review.
Author response: We appreciate this perspective. We removed CHIKV elements that were extraneous and generalized others relevant to both VEEV and CHIKV. Importantly, the sections highlighted by the reviewer involve studies that were focused almost exclusively on CHIKV with some VEEV evaluation done, as opposed to most other studies covered in this review that were intentionally developing compounds for V/W/EEEV. However, the drivers for SAR optimization and endpoints, including insights on CHIKV MOA, are relevant to medicinal chemists who are eyeing the scaffold for absence of toxicity across multiple cell lines, skeletal tunability for ADME and potential for V/W/EEEV. We also added a summary statement after the first section “In sum, these efforts may help guide additional work with this scaffold on encephalitic alphaviruses.”
Rev 4, item 8: In line 881 it is mentioned that it is important to pick relevant cell lines, I think that relevant virus strains should be added to this. There are many compounds that only test against TC83 which is a vaccine strain that is attenuated and can give false positives and failures when moving forward to WT strains. Many use TC83 because it is a BSL-2 strain so more labs can use this for initial test, but many times most of the work is done with TC83 and then one little figure at the end saying, oh and in a plaques assay it also works against VEEV Trd. Especially when looking at host directed responses, TC83 produces a vastly different inflammation profile than VEEV TrD or other WT strains.
Author response: Agreed. We have added content on this within the noted paragraph. “Considerations for early-stage compounds include an understanding of mechanism, demonstration of antiviral activity and selectivity indices in relevant cell lines, and with wild-type strains of alphaviruses that will likely be used in downstream efficacy models.”
Rev 4, item 9: Line 893-894 is a really important statement that should be expanded upon by at least a couple of sentences. The biggest gap in the small molecule antiviral field is the lack of compounds that make it into in vivo testing. There is lots of in vitro work but very little in vivo work, especially with WT strains. Apart of the expanding of this could be what are some of the difficulties that lead to less in vivo testing, lack of facilities, lack of ability to work with select agents, etc.
Author response: Agreed. This section has been revised as follows:
Drug discovery and development activities centered on EEVs are impeded by unique challenges, in addition to the traditional drug pipeline bottlenecks, risks, triage or failure observed during hit-to-lead, lead advancement and preclinical studies. While very early activities may avoid the use of highly pathogenic viruses or select agents, at some point, specialized biosafety facilities, regulatory oversight, and trained personnel are required to execute studies with relevant virus strains. VEEV and EEEV are categorized as select agents, except for the attenuated VEEV TC83 strain that can be used at a BSL2 level. Therefore, highly pathogenic, wild type strains of VEEV, WEEV, and EEEV are restricted to use within a limited number of BSL3 facilities. Additionally, our understanding of alphavirus pathogenesis and the intermediacy of host proteins continues to grow, hampered in part by the absence of X-ray crystal structures of targets such as the nsP4 viral polymerase that would help guide compound optimization. Priming the pipeline with high quality, drug-like chemical matter from the start is also imperative to enhance the success of translation. Data from validated animal models of EEV infection that strongly mirror the pathogenesis and hallmarks of the human disease and provide clear clinical endpoints is also critical. This point is especially poignant due to the applicability of the animal rule [187, 188], as patient recruitment for an EEV human clinical efficacy study is more difficult given the unpredictable nature of EEV outbreaks and coincident infections that obscure straightforward diagnoses of encephalitic alphavirus infections. “